# Symbolic Quantile Regression for the Interpretable Prediction of Conditional Quantiles

**Cas Oude Hoekstra**                                           *casoudehoekstra@gmail.com*
*Independent researcher, Amsterdam, the Netherlands*

**Floris den Hengst**                                           *f.den.hengst@vu.nl*
*Vrije Universiteit Amsterdam, the Netherlands*

**Reviewed on OpenReview:** *https://openreview.net/forum?id=x9OYbyPJOG*

## Abstract

Symbolic Regression (SR) is a well-established framework for generating interpretable or white-box predictive models. Although SR has been successfully applied to create interpretable estimates of the average of the outcome, it is currently not well understood how it can be used to estimate the relationship between variables at other points in the distribution of the target variable. Such estimates of e.g. the median or an extreme value provide a fuller picture of how predictive variables affect the outcome and are necessary in high-stakes, safety-critical application domains. This study introduces Symbolic Quantile Regression (SQR), an approach to predict conditional quantiles with SR. In an extensive evaluation, we find that SQR outperforms transparent models and performs comparably to a strong black-box baseline without compromising transparency. We also show how SQR can be used to explain differences in the target distribution by comparing models that predict extreme and central outcomes in an airline fuel usage case study. We conclude that SQR is suitable for predicting conditional quantiles and understanding interesting feature influences at varying quantiles.

## 1 Introduction

Symbolic regression (SR) offers an approach to uncover mathematical expressions that explain patterns in data. It captures these patterns in interpretable, closed-form expressions that can then be analyzed and interpreted. First, this is useful in the so-called discovery settings, where the study of the identified patterns increases the *understanding* of some phenomenon, as is the case in empirical science. Second, this is useful when making *predictions* in high-stakes domains, where accountability and safety are key considerations to the use of predictions in decision making (Bellemare et al., 2023). In general, SR has proven to be suitable for use cases that require understanding, mitigation of risks, and keeping in mind the broader goals of making predictions. It has therefore been applied in a wide range of fields, including astrophysics (Lemos et al., 2023), economics (Verstyuk & Douglas, 2022), medicine (Virgolin et al., 2020), mechanical engineering (Kronberger et al., 2018), chemistry (Hernandez et al., 2019), and others (Märtens & Izzo, 2022; Matsubara et al., 2022), as a result.

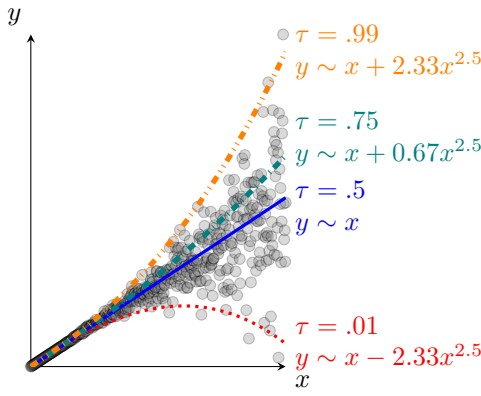

Figure 1: Conditional quantile functions at various quantile levels $\tau$ for a distribution where the variance of the target $y$ changes with the independent variable $x$.

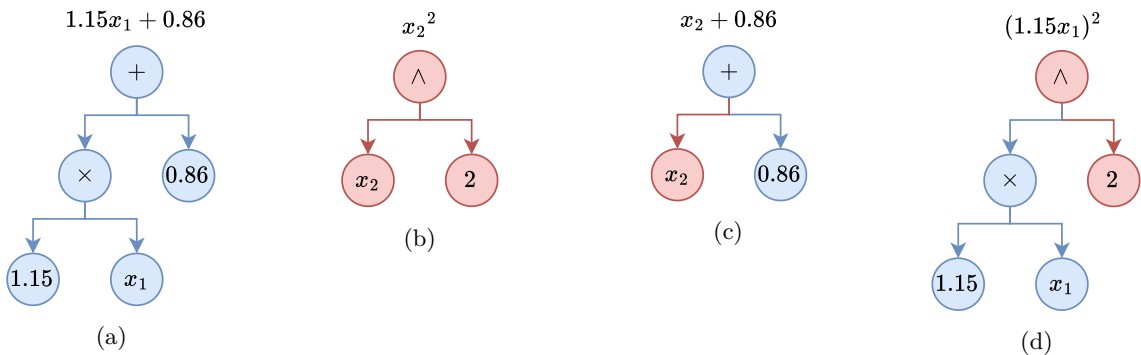

Figure 2: Two symbolic expression are represented as trees (2a and 2b), and combined through *crossover* to form two new expressions (2c and 2d).

Quantile regression (QR), on the other hand, focuses on making predictions at different locations of the outcome distribution by estimating different conditional quantile functions as visualized in Figure 1. Unlike traditional regression, which focuses on predicting a single central location of the outcome variable, QR accounts for variability and extremes, offering more robust predictions that can be used for better decision making (Koenker & Bassett Jr, 1978; Steinwart & Christmann, 2011). This is of paramount importance in contexts where data exhibit heteroskedasticity, such as in economics (Buchinsky, 1995), or when it is necessary to ensure that a certain proportion, say 90%, of actual values is lower than the predicted values, such as in survival analysis, reliability engineering, and healthcare (Hu et al., 2020; Koenker & Geling, 2001; Zheng et al., 2022).

However, current state-of-the-art QR models are black boxes and lack the interpretability required for high-stakes decision making that SR offers by design. This study therefore proposes a novel combination of SR and QR that provides interpretable predictive models for a given quantile level. Symbolic Quantile Regression (SQR) jointly optimizes for predictive performance and interpretability of the generated expressions by minimizing an established QR loss known as the pinball loss together with a loss for the interpretability of the expression.

We compare SQR with a number of state-of-the-art QR approaches in different quantiles in a substantial benchmark of 122 regression data sets. We assess its predictive performance and interpretability and find that SQR finds interpretable models while maintaining a predictive performance comparable to or better than state-of-the-art black-box models. We also present a case study from the commercial aviation domain, in which we apply SQR to a fuel consumption prediction problem to explore why some flights consume more fuel with the goal of reducing CO2 emissions. By comparing expressions that describe central and extreme fuel consumption levels, we find that higher velocities resulting from later departures explain extreme fuel usage. We thereby not only show that SQR is competitive in its predictive performance but also showcase how it can be used to create actionable insights on a real-world problem. Hence, SQR addresses key challenges related to the safe adoption of machine learning techniques in safety-critical and high-stakes domains by providing interpretable estimates of conditional quantile functions.

## 2 Background and Related Work

### 2.1 Symbolic Regression

Symbolic regression (SR) can be defined as a search process over the space of concise, closed-form mathematical expressions for an expression that best fits a dataset, thereby revealing the underlying patterns. For an i.i.d. data set $(X, y) \in \mathbb{X} \times \mathbb{R}$, where each input $x_i \in \mathbb{R}^d$ and output $y_i \in \mathbb{R}$, the goal of SR is to find a function $f : \mathbb{R}^d \to \mathbb{R}$ that accurately predicts the output given the input. In SR, the function $f$ represents a mathematical expression that captures the relationship between explanatory and target variables, and takes the form $f(x_i) = y_i + \epsilon$ for an error term $\epsilon$ (Koza, 1994).

Table 1: Complexity scores for operators.

| Token | Complexity |
|---|---|
| $+$, $-$, $\times$, feature, constant | 1 |
| $\div$, square | 2 |
| sin, cos | 3 |
| exp, log, $\sqrt{\cdot}$ | 4 |

The expression $f$ is modeled as a sequence of *tokens*, which include mathematical operations such as addition ($+$), subtraction (-), trigonometric functions (sin()), input features ($x_1 \ldots x_d$) and constants in $\mathbb{R}$. The search process aims to minimize a loss function, which is typically a measure of predictive performance, such as the mean squared error for regression tasks or the F1 score for binary classification, along with some measure of the interpretability of the expression (Visbeek et al., 2024). The interpretability of the expression is typically defined as its parsimony, i.e. the sum of the complexity scores assigned to each token, as illustrated in Table 1.

Historically, SR has been tackled using genetic programming (Koza, 1994), see Figure 2. Genetic programming is inspired by evolutionary biology and mimics natural selection through operations like crossover and mutation to evolve candidate solutions over generations. By iteratively combining and modifying instances in a population of solutions, it adheres to the principle of "survival of the fittest" by selecting models based on their performance according to a predefined loss or fitness function. The flexibility and robustness of population-based evolutionary methods make it a powerful tool for exploring the combinatorial solution space of mathematical expressions in SR.

Discovering an expression that is both predictive and interpretable not only enhances understanding of the phenomena underlying the data (the *discovery* use case) but also provides a reliable means for predicting the target variable (the *prediction* use case). In recent years, several extensions of the SR framework have emerged, including integrations with deep reinforcement learning and adaptations for classification tasks (Landajuela et al., 2022; Visbeek et al., 2024). Furthermore, the field of SR has been further developed through the introduction of various data sets and software tools that have advanced the research and practical applicability of SR (Orzechowski et al., 2018; Cranmer et al., 2020; La Cava et al., 2021).

## 2.2 Interpretability

Interpretability in machine learning has traditionally been evaluated using sparsity or simplicity metrics within a given model class (Jo et al., 2023). For instance, in linear models, interpretability is often measured by counting the number of nonzero coefficients, while in decision trees, interpretability is typically assessed by the number of nodes (i.e., branches and leaves). However, these metrics are limited because they do not allow meaningful comparisons across different model classes, as each class defines sparsity differently.

To address this limitation, the concept of decision complexity has been introduced as a more generalizable metric for sparsity that can be applied across various model classes. Decision complexity is defined as "the minimum number of parameters required for a classifier to make a prediction on a new data point" (Jo et al., 2023). This notion is generally referred to as parsimony in recent work and standardizes the measurement of interpretability by focusing on the essential components needed for decision making, regardless of the model type.

## 2.3 Quantile Regression

Quantile regression (QR), introduced by Koenker & Bassett Jr (1978), extends the concept of ordinary least squares (OLS) by estimating conditional quantiles of the target variable rather than its conditional mean.

The foundational method in this domain is Linear Quantile Regression (LQR) (Koenker & Bassett Jr, 1978). Unlike OLS, which minimizes the sum of squared residuals, LQR minimizes the sum of asymmetrically weighted absolute residuals, known as the pinball loss (or tilted absolute value function, see equation 2).

While LQR is highly interpretable and computationally efficient, it assumes a linear relationship between the covariates and the conditional quantiles, which limits its flexibility in modeling complex, non-linear real-world phenomena (Koenker & Hallock, 2001).

To address the limitations of linearity, non-parametric approaches such as Quantile Decision Trees (QDT) and ensemble methods have been developed. QDTs adapt the standard decision tree algorithm by modifying the splitting criteria or leaf node estimation to target specific quantiles. For instance, Chaudhuri & Loh (2002) proposed to use quantile regression in the nodes of the tree. A more common adaptation involves growing trees to minimize the deviation from the target quantile or estimating the empirical quantile distribution within each leaf (Meinshausen & Ridgeway, 2006).

Gradient boosting frameworks have further advanced the state-of-the-art in predictive performance (Friedman, 2001). LightGBM (Light Gradient Boosting Machine) by Ke et al. (2017) is a widely used implementation that supports quantile regression. It constructs an ensemble of decision trees by iteratively minimizing the quantile loss function using gradient descent. While black-box boosting machine-based models often outperform linear baselines in terms of predictive performance, they lack the intrinsic transparency required for high-stakes decision-making, as the resulting models are ensembles of hundreds or thousands of decision trees.

## 3 Symbolic Quantile Regression

We now turn to the main technical contribution, which is the extension of SR so that it can be used to predict various locations of the target distributions while maintaining interpretability. We formalize this problem as follows. Let $(\mathcal{X}, \mathcal{Y})$ be random variables where $\mathcal{X} \in \mathbb{R}^d$ represents a $d$ dimensional input and $\mathcal{Y} \in \mathbb{R}$ the response variable or target. Our goal is to estimate the conditional quantile function $Q_{\mathcal{Y}}(\tau|\mathcal{X} = X)$ for a specified quantile level $\tau \in (0, 1)$:

$$Q_{\mathcal{Y}}(\tau|\mathcal{X} = X) := \inf \{q \in \mathbb{R} : \mathbb{P}(\mathcal{Y} \leq q|\mathcal{X} = X) \geq \tau\}. \tag{1}$$

We assume the presence of a data set of i.i.d. samples $D := (X, y)^n \sim (\mathcal{X}, \mathcal{Y})$ of size $n$.

### 3.1 Estimating Conditional Quantiles

A common approach to estimating the conditional quantile function $Q_{\mathcal{Y}}(\tau|\mathcal{X})$ given a dataset is to use empirical risk minimization with the use of the pinball loss. This loss function resembles an asymmetric absolute value function and penalizes underestimation and overestimation differently, making it well-suited for quantile estimation tasks where such asymmetries are meaningful. Specifically, for a given error $\varepsilon_i := y_i - f(X_i)$ where $f(X_i)$ is the predicted value, the pinball loss is defined as:

$$L_\tau(\varepsilon_i) := \begin{cases} \tau(\varepsilon_i) & \text{if } \varepsilon_i \geq 0, \\ (\tau - 1)(\varepsilon_i) & \text{if } \varepsilon_i < 0 \end{cases} \tag{2}$$

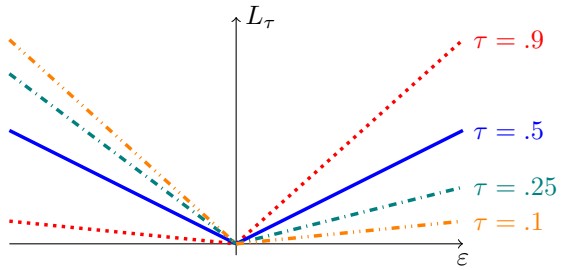

Figure 3: Pinball loss function for various values of conditional quantile $\tau$. Errors $\varepsilon$ are penalized asymmetrically for $\tau \neq 0.5$ to predict values close to the desired $\tau$.

Figure 3 visualizes this function for different quantile levels $\tau$. For example, with $\tau = .9$, an underestimate $\varepsilon = -1$ is penalized nine times more than its corresponding overestimate $\varepsilon = 1$. As a result, the optimization is biased toward over-prediction, yielding estimates that align with the 90th empirical conditional quantile.

SQR estimates the conditional quantiles of a distribution $P$ with empirical risk minimization. Let $P$ be a distribution in $\mathbb{X} \times \mathbb{R}$, with $\mathbb{X}$ being an arbitrary set equipped with a $\sigma$ parameter. Here, $\mathbb{R}$ represents the target space and $\mathbb{X}$ the input feature space. That is, each $X \in \mathbb{X}$ corresponds to a set of predictor variables and each $y \in \mathbb{R}$ corresponds to an observed response variable. The conditional quantile function $F^*_{\tau, P}(X)$

for a given input $X \in \mathbb{X}$ is defined as:

$$F_{\tau,P}^*(X) := \{t \in \mathbb{R} : \mathbb{P}(y \leq t | \mathcal{X} = X) \geq \tau \text{ and } \mathbb{P}(y \geq t | \mathcal{X} = X) \geq 1 - \tau\} \tag{3}$$

This function identifies the threshold $t$ such that the probability of observing a target value $y$ below $t$ (given $\mathcal{X} = X$) is at least $\tau$, and the probability of observing a value above this threshold is at least $1 - \tau$. The associated expected risk $R$ for a predictor $f : \mathbb{X} \to \mathbb{R}$ is:

$$R_{L,P}(f) = \mathbb{E}_{(X,y) \sim P}[L_\tau(y, f(X))] = \int_{\mathbb{X} \times \mathbb{R}} L_\tau(y, f(X)) dP(X, y) \tag{4}$$

and the objective is to find $f_{\tau,P}^*$ that minimizes this risk to effectively balance underestimation and overestimation for the desired quantile level.

However, while empirical risk minimization provides a principled means to estimate quantiles from data, it is fundamentally based on the quality and representativeness of the observed sample. As evident in equation 4, minimizing the empirical risk does not guarantee that the resulting model perfectly captures the true conditional quantile. This issue is particularly pronounced in regions of the distribution where data is sparse or noisy, such as the tails, which are often the focus in high-stakes settings.

Crucially, this challenge is not unique to SQR or the use of pinball loss. Rather, this is a fundamental limitation that arises in any approach to quantile estimation from finite data. As such, careful evaluation and validation are essential to ensure robust quantile estimation, and our experiments are designed with these considerations in mind.

### 3.2 Optimizing for Interpretability

Having established a suitable loss for estimating the conditional quantile function, we now turn to the objective of interpretability. We operationalize interpretability as *parsimony*, defined as a preference for concise symbolic expressions with minimal structural and functional complexity. Parsimony is particularly suitable in high stakes settings, where trust and safety are paramount, as simpler models are more amenable to human inspection, validation, and understanding, thus promoting trust and perceived trustworthiness (Bansal et al., 2019).

We define parsimony following established conventions (Petersen et al., 2021; 2020) For an expression $f$, composed of tokens $t$ with associated token complexity scores $|t|$, as detailed in Table 1, its parsimony $|f|$ is defined as:

$$|f| := \sum_{t \in f} |t|. \tag{5}$$

Since we may not know what level of interpretability is necessary and attainable, we construct a Pareto front (PF) of the best expressions across a range of parsimony levels, from which a single expression with optimal predictive performance and acceptable interpretability can be selected. A preferred solution can then be selected from this front based on, e.g., the elbow method or on requirements from the use case or domain at hand (Thorndike, 1953).

Formally, the optimization problem for SQR is therefore:

$$\text{PF}_\tau(\mathbb{C}, D) := \left\{ \arg\min_{f \in \mathcal{F}} \left( \sum_{i=1}^n L_\tau(\varepsilon_i) \right) \middle| |f| = c, \ c \in \mathbb{C} \right\} \tag{6}$$

where $\mathcal{F}$ denotes the space of all possible expressions that can be constructed from the token library and $\mathbb{C} \subset \mathbb{N}_{[1, C_{\max}]}$ a set of possible parsimony scores under consideration.

### 3.3 Implementation

We propose an implementation of SQR based on PySR, a symbolic regression optimization engine that combines evolutionary search with local optimization and adaptive regularization by Cranmer (2023). We

use this engine due to its support for these features, its strong empirical performance in noisy conditions, and ease in implementation of research prototypes.

Our implementation addresses the optimization problem described in equation 6 by maintaining a global registry of best solutions per complexity level known as the *hall of fame*. Specifically, a candidate best solution for every integer complexity level $c \in \mathbb{C}$ under consideration is stored in this registry. As the evolutionary search proceeds, every generated candidate $\hat{f}'$ is evaluated against the current best solution with the same complexity. If the new candidate $\hat{f}'$ achieves a lower pinball loss $L_\tau$ according to equation 2 than the current best candidate for its complexity level, the new candidate replaces the current one in the hall of fame. At the conclusion of the search, the hall of fame constitutes an approximation to problem equation 6. The final model is selected directly from this Pareto front using the elbow method in our empirical evaluation.

The evolutionary process consists of an outer and an inner evolutionary process. In the outer process, several independent populations of symbolic expressions are evolved for a preset number of iterations. Their independence is to reduce the risk of the process becoming trapped in a local optimum. At the end of a prespecified number of iterations, solutions in each population are stochastically replaced by solutions from independent populations to improve coverage of the entire search space. This process continues until a prespecified number of iterations have been reached.

In the inner evolutionary process, each population undergoes repeated cycles of parent selection, evolution, simplification, constant optimization, and survivor selection as visualized in Figure 4. Parents are selected from a random sample based on tournament selection. The selected parent expressions are modified via mutation and crossover operators to explore new structural forms, simplified via algebraic rewriting to reduce complexity and improve interpretability, and its scalar constants are fine-tuned using a local numerical optimization approach known as BFGS (Nocedal & Wright, 2006).

The resulting candidate solutions are added to the population stochastically proportional to their *fitness* score $g(\hat{f})$) defined as follows:

$$g(\hat{f}) := L_\tau\left(y, \hat{f}(\boldsymbol{X})\right) + \lambda\left(|\hat{f}|\right)|\hat{f}|$$

where $L_t$ according to equation 2, $|\cdot|$ according to equation 5, and $\lambda(\cdot)$ an adaptive parsimony coefficient that changes over time based on the frequency distribution of complexity sizes in the current population. This $\lambda$ term ensures that if some complexity level $c \in \mathbb{C}$ is overrepresented in the current population, the fitness for all candidates $\hat{f}$ such that $|\hat{f}| = c$ is reduced to correct for this over time. This fitness score serves as the energy state for a simulated annealing process that defines the acceptance of a candidate $\hat{f}'$ over the oldest member in the population $\hat{f}$ (Real et al., 2019):

$$\mathbb{P}(\text{accept } \hat{f}' \text{ over } \hat{f}) \sim \exp\left(-\frac{g(\hat{f}) - g(\hat{f}')}{\alpha T}\right)$$

where $\alpha \in [0, 1]$ a hyperparameter to scale the fitness scores with respect to the temperature parameter $T \in [0, 1]$.

This combination of evolutionary search, local optimization, age-aware survival selection, and adaptive complexity control allows symbolic regression to remain effective in finding a diverse set of solutions under high noise across the range of complexity scores under consideration, making it well suited for modeling quantile functions where interpretability and robustness are critical.

## 4    Experiments

We evaluate SQR empirically on a substantive benchmark consisting of 122 datasets of varying dimensionality, and compare it to existing black-box and transparent models for QR. We focus on the predictions in a central and extreme location of the target variable with $\tau = .5, \tau = .9$, respectively, to assess the performance in typical and extreme conditions. See the appendices for all details about the datasets, statistical testing, and additional results.

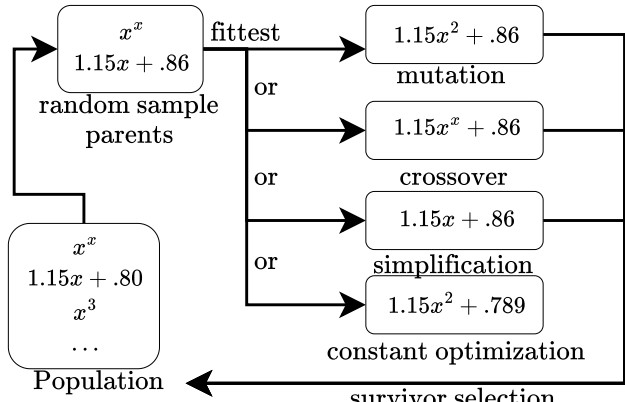

Figure 4: Evolutionary optimization process, adopted from Cranmer (2023).

### 4.1 Datasets

SRBench (La Cava et al., 2021) is the de facto standard for evaluating SR approaches. It comprises 252 synthetic and real-world data sets with and without noise in the target variable. We exclude all data sets without noise in the target variable for our purposes, because the conditional quantiles for a target without noise all lie at the same location. The noisy condition is also more realistic. This leaves an evaluation on 122 real-world and synthetic datasets of varying dimensionality and size.

### 4.2 Baselines

A set of models with state-of-the-art predictive performance and varying degrees of interpretability and parsimony were selected. Both transparent and black-box models were included to highlight the strengths and weaknesses of each approach in the context of quantile regression.

**Linear Quantile Regressor (LQR)** by Seabold & Perktold (2010) is widely adopted for its simplicity and interpretability. LQR cannot capture nonlinearities, but is robust to overfitting and generally considered highly transparent (Yu et al., 2003).

**Quantile Decision Tree (QDT)** by Pedregosa et al. (2011) is regarded as a transparent model, but may suffer from poor performance on small datasets, may overfit, and sometimes struggles with modeling linearities (Loh, 2014).

**LightGBM Quantile Regressor (LGBM)** by Ke et al. (2017) is selected for its ability to capture complex relationships in large data sets with competitive computation characteristics (Bentéjac et al., 2021). Since LGBM provides black-box models, it serves as a high-performance, albeit non-transparent, model.

**SQR** is our approach trained on the full training data set.

**SQR10k** is our approach trained on a random sample of at most 10k train instances. For datasets smaller than 10k, we train without sampling. We include this version of the approach to assess the effect of limited data when aiming for a model with high interpretability.

SQR was implemented and evaluated with the PySR software and default hyperparameters were used. (Cranmer et al., 2020). [1] The hyperparameters of LGBM, QDT, and LQR were optimized per dataset using Optuna and five-fold cross validation (Akiba et al., 2019). See Appendices D and F for all used settings, hyperparameters and implementation details.

---

[1] https://github.com/florisdenhengst/SQR

### 4.3 Evaluation

We include metrics based on best practices and recommendations from the literature (Steinwart & Christ-mann, 2011; Chung et al., 2021), and use a combination of metrics to ensure that our evaluations are valid given the challenges of estimating conditional quantiles on finite samples. See Appendix A for a detailed discussion on the issues associated with using only the pinball loss for evaluation.

Firstly, we employ a normalized version of the pinball loss that enables comparisons across datasets and is known as the quantile loss. This measure averages the loss in equation 2 in all instances of the test set. This total average loss is then normalized to $[0, 1]$ using the range of the target variable to ensure that data sets with target variables with a large range do not have an outsized influence on the evaluation. We define the Normalized Quantile Loss (nql) $\text{nql} := \frac{\frac{1}{n}\sum_{i=1}^{n} L_\tau(\boldsymbol{y}_i, f(\boldsymbol{X}_{i,:}))}{\max(\boldsymbol{y}) - \min(\boldsymbol{y})}$ for all $n$ items in some test set.

We complement the normalized quantile loss with a measure that expresses whether predictions align well with the specified quantile level $\tau$, known as the absolute coverage error:

$$\text{Absolute Coverage Error (ace)} := |\text{Cov}(\tau) - \tau| \tag{7}$$

for an empirical coverage defined as:

$$\text{Cov}(\tau, D) := \frac{1}{n} \sum_{i=1}^{n} 1\left(y_i \leq f(X_i)\right). \tag{8}$$

For example, when $\tau = .9$, the empirical coverage ideally reflects a 90% proportion of the test data that falls below the predicted quantile.

Additionally, we include model parsimony as a measure to capture the notion of interpretability. Since we are interested in comparing the parsimony of various models across datasets, we opted to use average parsimony across models and datasets as defined in equation 5. We define the parsimony for baselines as usual in literature, see Appendix B for details. We report run times to assess the practical feasibility of our approach. In doing so, we note that this comparison is between a research prototype implementation for SQR, against highly optimized industry-grade implementations of the baselines.

We additionally evaluate the robustness of the approach in two additional experiments. In the first, the quantile levels are varied $\tau \in (.5, .6, .7, .8)$. In the second, we evaluate the models in an out-of-distribution (OOD) scenario. Here we select the best performing model (LGBM) and train it on each dataset. We calculate the feature importances using Shapley values for this model on each dataset. We then use the most predictive feature per dataset to split the data into a train and a test split. Specifically, the split is made at the 90th quantile of this most informative feature where the largest split is used for training and the smallest forms the out-of-distribution test split. To limit the computational burden, we run these additional experiments on a random sample of 10k data points.

Five-fold cross-validation was used across experiments, resulting in five different test scores for every measure and across quantiles. These scores were averaged per dataset, resulting in two scores for each measure per model for every data set. We present the average (mean) and standard deviation (SD) over all data sets across results. For the main experiments, nonparametric statistical tests were used due to nonnormality of the data according to the Shapiro-Wilk test ($p < 0.05$) for most results. For comparisons between measures and quantiles, the Friedman test was used. Significant results were further analyzed using the Paired Wilcoxon signed rank test. Bonferroni correction was applied in all cases of multiple comparisons.

### 4.4 Results

Table 2 lists the predictive performance metrics and shows that SQR is the transparent model that performs best. The results are consistent at both quantile levels $\tau = .5$ and $\tau = .9$. This table further shows that SQR achieves this strong performance at a lower cost in terms of parsimony than transparent alternatives.

To further evaluate the performance of interpretable models in the three selected metrics in Table 2, we performed statistical significance tests to assess differences per (metric, quantile level)-pair for all transparent

Table 2: Comparison across normalized quantile loss, absolute coverage error, parsimony, and transparency. **Bold** indicates significant best among transparent models.

| | Normalized Quantile Loss | | Absolute Coverage Error | | Parsimony | | Transparent |
|---|---|---|---|---|---|---|---|
| | $\tau = .5$ | $\tau = .9$ | $\tau = .5$ | $\tau = .9$ | $\tau = .5$ | $\tau = .9$ | |
| LGBM | $.042 \pm .022$ | $.025 \pm .012$ | $.064 \pm .070$ | $.039 \pm .038$ | $-$ | $-$ | |
| LQR | $.059 \pm .030$ | $.059 \pm .044$ | $\mathbf{.065} \pm .081$ | $.287 \pm .207$ | $19.54 \pm 21.14$ | $19.57 \pm 21.04$ | ✓ |
| QDT | $.039 \pm .017$ | $.020 \pm .001$ | $.088 \pm .10$ | $.066 \pm .057$ | $323.46 \pm 1306.47$ | $210.82 \pm 1080.43$ | ✓ |
| SQR | $\mathbf{.030} \pm .021$ | $\mathbf{.015} \pm .013$ | $.082 \pm .092$ | $\mathbf{.049} \pm .061$ | $\mathbf{10.04} \pm 4.09$ | $\mathbf{9.85} \pm 3.67$ | ✓ |
| SQR10K | $.033 \pm .022$ | $.017 \pm .013$ | $.084 \pm .084$ | $.058 \pm .070$ | $9.82 \pm 4.40$ | $9.71 \pm 4.22$ | ✓ |

Table 3: Average run time (ms).

| | $\tau = .5$ | $\tau = .9$ |
|---|---|---|
| LGBM | $5.01 \pm 46.61$ | $9.81 \pm 94.02$ |
| LQR | $10.31 \pm 32.36$ | $11.51 \pm 37.87$ |
| QDT | $0.32 \pm 0.68$ | $0.35 \pm 0.77$ |
| SQR | $3299.12 \pm 11652.14$ | $1906.94 \pm 5757.59$ |
| SQR10K | $166.77 \pm 171.50$ | $166.36 \pm 167.87$ |

models. Each test's null hypothesis states that there are no significant differences between two models for a specific metric-quantile level pair, following this template:

$H_{\text{null}}^{\mathbf{metric},\tau}$ The differences in 'metric' between models at the $\tau$ quantile level are not significant, and model A cannot be said to significantly outperform model B or vice versa.

A two-stage analysis approach was employed to reduce the number of statistical tests. First, significant differences between results were assessed per (metric, quantile level)-pair. If differences were significant, we evaluated which model was the significant best.

Statistical significance at the metric-quantile level was first assessed using a Bonferroni corrected Friedman test per pair at the metric-quantile level. Given three metrics and two quantile levels, the corrected alpha level was set to $\alpha = \frac{0.05}{6} \approx 0.00833$. Significant differences were observed at both quantiles, for all of normalized quantile loss, absolute coverage error and parsimony.

A second stage of statistical tests was used to assess *which* of the models significantly outperform the others using pairwise comparisons in a Bonferroni-corrected Wilcoxon signed rank test. With 18 pairwise comparisons in total, the corrected alpha level was established at $\alpha = \frac{0.05}{18} \approx 0.00278$. The performance improvements of SQR over other transparent models were significant for all metrics and conditions, with the exception of the improvement of SQR over QDT in absolute coverage error for $\tau = .5$.

Considering all statistical tests on the results in Table 2 together, we summarize the evaluation for all hypotheses as follows (see Appendix C for all results). Significant differences in normalized quantile loss were observed. SQR consistently outperformed both LQR and QDT, achieving the lowest loss across both tested quantiles. Significant differences were also observed for absolute coverage error, with LQR demonstrating the best calibration. Regarding parsimony, differences were significant in all configurations, with SQR performing best for both quantile levels. These results indicate that SQR generally outperforms the other interpretable models in both predictive performance and interpretability when trained on the full training set.

Looking at the runtime results in Table 3, we find that the prototype implementation of SQR comes at substantial additional computational cost. However, this cost can be significantly brought down through sampling as the run times for SQR10K show: these significantly bring down the run times at limited cost in predictive performance or parsimony. Furthermore, we believe that the runtime of our approach can be

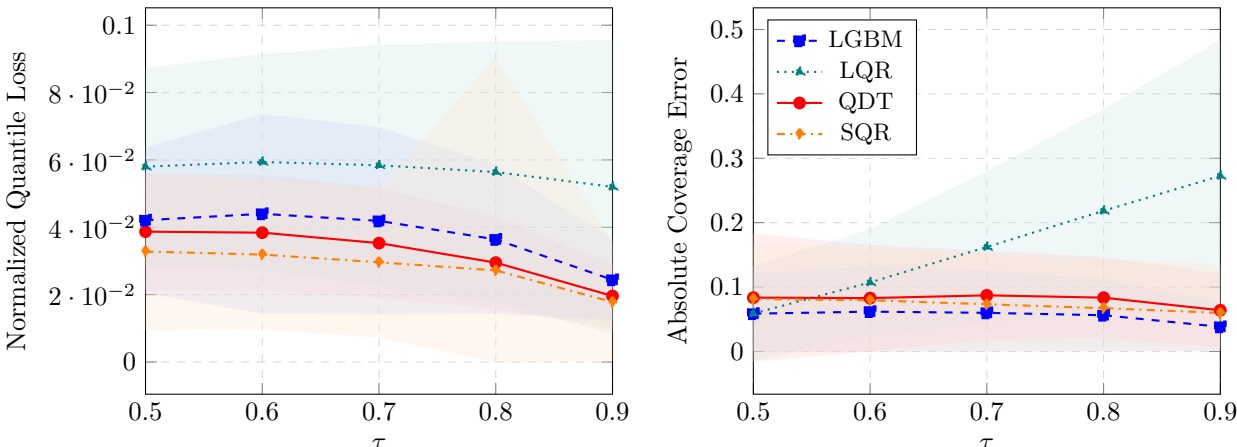

Figure 5: Result for varying target quantiles $\tau$.

Table 4: Out-of-distribution (OOD) results. OOD data was generated by removing 90th quantile data for the most predictive feature for LGBM. $\Delta$ columns indicate performance difference with main results in Table 2, **bold** indicates significant differences for transparent models.

| Model | Normalized Quantile Loss | | $\Delta$ Loss (%) | | Absolute Coverage Error | | $\Delta$ Coverage (%) | | Transp. |
|---|---|---|---|---|---|---|---|---|---|
| | $\tau = .5$ | $\tau = .9$ | $\tau = .5$ | $\tau = .9$ | $\tau = .5$ | $\tau = .9$ | $\tau = .5$ | $\tau = .9$ | |
| LGBM | $.085 \pm .045$ | $.044 \pm .051$ | $+102\%$ | $+76\%$ | $.355 \pm .138$ | $.260 \pm .209$ | $+454\%$ | $+566\%$ | |
| LQR | $.102 \pm .097$ | $.163 \pm .141$ | $\mathbf{+73}\%$ | $+176\%$ | $.314 \pm .165$ | $.563 \pm .333$ | $+383\%$ | $\mathbf{+96}\%$ | ✓ |
| QDT | $.072 \pm .042$ | $.045 \pm .049$ | $+84\%$ | $+125\%$ | $.309 \pm .152$ | $.297 \pm .232$ | $+251\%$ | $+350\%$ | ✓ |
| SQR | $.060 \pm .061$ | $\mathbf{.030} \pm .077$ | $+100\%$ | $\mathbf{+100}\%$ | $\mathbf{.244} \pm .169$ | $\mathbf{.192} \pm .203$ | $\mathbf{+197}\%$ | $+291\%$ | ✓ |

reduced by two orders of magnitude based on a recent comparison of the underlying optimization framework with a highly optimized implementation (Tonda, 2025).

A detailed examination of the results in Appendix E provides further context for the performance differences: QDTs tend to scale parsimony with the number of samples $n$ whereas LQR scales strictly with data dimensionality $d$. In contrast, SQR generally maintains low parsimony scores (often $< 15$) and exhibits a relatively stable parsimony across datasets with varying characteristics.

SQRs explicit preference for low-parsimony solutions appears to act as an explicit regularizer, which also affects SQR's performance profile when looking at predictive performance metrics quantile loss and absolute coverage error. In high-dimensional, low-sample regimes, SQR may better isolate a small number of features, whereas LQR is more susceptible to overfitting, occasionally resulting in higher Absolute Coverage Errors at extreme quantiles. Conversely, on datasets with a large number of observations, SQR's preference for low-capacity models can become a limitation. In these large $n$, low $d$ scenarios, SQR may underfit compared to QDT or LGBM, which can more flexibly increase their capacity to model complex, non-linear relationships.

Figure 5 shows the performance across varying quantile levels. It indicates that the performance across $\tau$ levels is consistent with those reported in the main experiments. Table 4 displays the OOD results. We first note that all methods incur a penalty in predictive performance when comparing to the main results in Table 2. Second, we note that SQR maintains its position as overall significantly best performing model. Third, we find that SQR outperforms LGBM in all metrics and target quantiles. This may be explained by the preference for simpler models in SQR, which acts as an explicit regularizer to avoid overfitting and help generalize to OOD regions of the data. Since LGBM was used to obtain the OOD split, we opt to remain cautious here as this may have caused a disproportionate performance penalty for this model.

Table 5: Features used for predicting and explaining airline fuel usage.

| Abbreviation | Name | Description |
|---|---|---|
| ASF | Adjusted Speeding Factor | Speeding due to departure delays. |
| GCD | Great Circle Distance | Shortest distance between departure and arrival airports (km). |
| AWC | Average Wind Component | Average wind component relative to the aircraft's direction. |
| TP | Total Pax | Total number of passengers on the flight. |

## 5 Understanding Extreme Airplane Fuel Use

We applied SQR to a real-world use case to assess its practical utility for creating interpretable predictions at different locations of the target variable. In this case study, the first objective is to predict extreme quantiles to ensure sufficient fuel is loaded. Interpretability is a prerequisite here due to industry regulations which require transparent models that are subject to human oversight. The second objective is to explain differences between central and extreme fuel usage in order to reduce fuel consumption and CO2 emissions.

To pursue the initial goal of accurate predictions, we use SQR to create expressions that capture extreme ($\tau = .9$) fuel consumption across flights between two locations. To pursue the second goal of understanding differences between central and extreme fuel usage, we additionally create expressions for the central conditional quantile ($\tau = .5$) function. By comparing the resulting expressions we gain understanding of why fuel usage may be high for some flights in comparison to others, demonstrating SQR's potential to uncover actionable insights by predicting patterns in the data at different quantiles.

### 5.1 Dataset

The dataset for this use case contains flights of the Boeing 777 aircraft operated by an internationally operating airline company. The target variable is the amount of fuel consumed during flight in kg, all explanatory variables are listed in Table 5. The adjusted speeding factor (ASF) is a concept that captures pilot speeding behavior. It is defined based on scheduled and actual departure times and flight duration. If the plane departed earlier than scheduled and the actual flight duration is equal to or longer than the planned flight duration, then ASF is the ratio of actual to planned flight duration. If the plane departs later than scheduled and the actual flight duration is shorter than the planned duration, ASF is the ratio of planned to actual duration. In all other cases, ASF is set to 1.0, indicating no speeding adjustment. A selection was made of flights on two days of the week for a duration of four years (2019-2023) to ensure a sufficient size and to combat the effects of airport congestion, passenger load, and weather conditions.

### 5.2 Results

We now present, analyze and compare the expressions for the 50th and 90th quantiles in order to explain extreme airline fuel usage. We highlight how ASF, a variable capturing speeding, impacts fuel consumption differently under median and extreme conditions.

For the 50th quantile, the selected expression produces an empirical coverage of 0.51, and a quantile loss of 1746. This corresponds to a normalized quantile loss of 0.03 after approximate normalization [2]:

$$\hat{f}_{\tau=50} = 7.216 \times GCD + 0.003 \times GCD \times (TP + 0.045 \times GCD - 7.22 \times AWC) + 1676.6 \qquad (9)$$

The expression suggests a linear relationship primarily driven by the traveled distance (GCD), with interactions involving the number of passengers (TP) and wind conditions (AWC). The coverage is close to the target quantile level.

The 90th quantile expression produced an empirical coverage of 0.91, which is close to the target quantile of 0.9. The unnormalized quantile loss sits at 944.59, which corresponds to an approximate normalized quantile

---

[2]we approximate the normalization constants for minimum and maximum fuel usage minimum and maximum values to avoid disclosure of sensitive data at a conservative 12000kg and 80000kg respectively based on (Kühn & Scholz, 2023).

loss of 0.01, again indicating a good fit with the data:

$$\hat{f}_{\tau=90} = 2360.4 \times ASF^2 + \frac{GCD \times (TP + 2126.03)}{238.2 \times ASF + 0.45 \times AWC} \tag{10}$$

The expression suggests complex interactions in which the speeding component (ASF) dominates other factors such as travel distance (GCD), passenger load (TP) and wind conditions (AWC). The quadratic ASF term is additive to the GCD term. We hypothesize that the quadratic ASF term captures the non-linear increase in drag associated with high-thrust speeding maneuvers such as rapid climbs or descents, which act as distinct operational costs independent of the standard cruise efficiency.

The expressions reveal remarkable differences in the impact of speeding (ASF) on fuel consumption. For the 50th quantile, fuel consumption is primarily driven by distance, with moderate effects from passenger load and wind conditions. In contrast, the 90th quantile shows that speeding plays a crucial role and indicates that speeding leads to extreme fuel consumption. Hence, a decrease in speeding is expected to decrease extreme fuel consumption. This insight may be used by airlines that aim to reduce fuel consumption and CO2 emissions, for instance by focusing efforts on a timely departure or a schedule with planned arrival times that accommodate for departure delays.

## 6 Discussion

We introduced Symbolic Quantile Regression (SQR), a novel method that combines Symbolic Regression (SR) with Quantile Regression (QR) to produce interpretable predictors of conditional quantiles. In an extensive benchmark, SQR demonstrated competitive accuracy at both the median (50th) and upper (90th) quantiles, outperforming related methods across predictive performance, empirical calibration, and interpretability metrics at a substantial increase in computational cost in its current implementation.

SQR balances predictive accuracy with model transparency, a key requirement in domains where interpretability is not just desirable but essential. Our results suggest that SQR is effective in estimating quantiles with concise symbolic models, making it particularly suitable for applications in high-stakes settings such as healthcare, finance, engineering, and scientific discovery. In these contexts, *understanding* the conditions under which certain outcomes arise is as important as the *predictions* of outcomes themselves. While fully uncovering the phenomena is a causal problem which may require counterfactual data and/or further assumptions to address fully automatically, our SQR approach can enhance understanding of the phenomena at hand based on a given set of data.

However, SQRs built-in capacity restrictions may result in SQR not fully leveraging the information available in low-dimensional large datasets. In settings with tens of thousands of observations, SQR's predictive performance often lags behind that of tree-based models, which more flexibly adapt their complexity with the data set size. Consequently, SQR is likely best suited for applications where interpretability, regularization, and robust prediction are prioritized over maximizing predictive performance in large datasets.

A real-world case study in the airline sector further illustrated the practical utility of SQR. By modeling fuel consumption across quantiles, SQR revealed the impact of speeding on high fuel usage. This insight can inform operational strategies to reduce both costs and environmental impact. The strong empirical coverage of the model in extreme conditions and its interpretability underscore its potential for integration into daily decision-making pipelines in the high-stakes domain of commercial aviation.

Despite these promising results, several limitations merit discussion. Our token-based complexity metric to operationalize interpretability, while standard and scalable across datasets, remains a proxy. Future work should explore adaptive and human-centered metrics, potentially learned in human-in-the-loop settings to better capture domain- and user-specific notions of simplicity and relevance (Nadizar et al., 2024).

Another possibility for future research lies in the extension of SQR to calibrated prediction intervals, combining it with methods such as conformal prediction (Fontana et al., 2023) or joint optimization frameworks for interval predictors (Soares & Fagundes, 2018). These directions would extend and complement SQR to further enhance the reliability of symbolic predictive models, under e.g. distributional shift or uncertainty.

In conclusion, SQR delivers accurate and interpretable quantile estimates with empirical robustness and demonstrated real-world relevance. It advances the field's broader goals of safe, explainable and actionable AI by supporting decision-making with accurate *predictions*, and by providing an *understanding* of the underlying phenomena. With its unique combination of modeling various locations of the target distribution and the usage of symbolic functions to capture complex relationships concisely, SQR represents a compelling step in, and useful tool for, transparent and robust machine learning.

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

## A    Quantile Dependence of the Pinball Loss

To illustrate the potential quantile dependence of the pinball loss consider the following example:

For 100 predictions at the 50th quantile, one expects 50 predictions to be below the target and 50 above the target, resulting in a simplified mean pinball loss of:

$$\text{Mean Pinball Loss}_{50th} = \frac{(50 \times 0.5) + (50 \times 0.5)}{100} = 0.5$$

For 100 predictions at the 90th quantile, one expects 10 predictions to be below the target and 90 above the target, leading to:

$$\text{Mean Pinball Loss}_{90th} = \frac{(10 \times 0.9) + (90 \times 0.1)}{100} = 0.18$$

Although this calculation does not account for the extent to which the model overpredicts or underpredicts, it suggests that the pinball loss at the 90th quantile is generally lower than the pinball loss at the 50th

quantile, irrespective of the predictive power. This hypothesis is further supported by the quantitative results presented in this thesis.

The asymmetrical nature of the pinball loss metric indicates its dependence on the chosen quantile, which poses challenges for consistent evaluation across different quantiles.

## B  Model Parsimony Details

Table 6: Model Parsimony Weights

| Token | Complexity |
|---|---|
| **SQR Complexity** | |
| $+, -, \times, \text{Feature}, \text{Constant}$ | 1 |
| $\div, \text{square}$ | 2 |
| $\sin, \cos$ | 3 |
| $\exp, \log, \sqrt{\cdot}$ | 4 |
| **LQR Complexity** | |
| Feature, Bias | 1 |
| **QDT Complexity** | |
| Node | 1 |

## C  Statistical Testing

For the 50th quantile:

- **H0$_{1,50}$:** SQR does not significantly outperform other quantile regressors in terms of Normalized Pinball Loss on the benchmark dataset.

- **H1$_{1,50}$:** SQR significantly outperforms other quantile regressors in terms of Normalized Pinball Loss on the benchmark dataset.

- **H0$_{2,50}$:** SQR does not significantly outperform other quantile regressors in terms of empirical coverage (Absolute Coverage Error) on the benchmark dataset.

- **H1$_{2,50}$:** SQR significantly outperforms other quantile regressors in terms of empirical coverage (Absolute Coverage Error) on the benchmark dataset.

- **H0$_{3,50}$:** SQR does not significantly outperform other quantile regressors in terms of Model parsimony on the benchmark dataset.

Table 7: Significance of results with Bonferroni-corrected Friedman tests.

| Metric | Abbr. | $\tau$ | Test statistic | p-value | significant |
|---|---|---|---|---|---|
| Normalized quantile loss | nql | 0.5 | 311.63 | $< 0.001$ | **Yes** |
| Absolute coverage error | ace | 0.5 | 18.67 | $< 0.001$ | **Yes** |
| Parsimony | c | 0.5 | 516.76 | $< 0.001$ | **Yes** |
| Normalized quantile loss | nql | 0.9 | 495.45 | $< 0.001$ | **Yes** |
| Absolute coverage error | ace | 0.9 | 302.63 | $< 0.001$ | **Yes** |
| Parsimony | c | 0.9 | 242.86 | $< 0.001$ | **Yes** |

Table 8: Pairwise Wilcoxon Signed-Rank Test Results for $\tau = 0.5$.

| Metric | Abbr. | Models | $T$-statistic | p-value | Significant |
|---|---|---|---|---|---|
| Normalized quantile loss | nql | SQR vs QDT | 35440.0 | $< 0.001$ | **Yes** |
| | | SQR vs LQR | 12151.0 | $< 0.001$ | **Yes** |
| | | QDT vs LQR | 21771.0 | $< 0.001$ | **Yes** |
| Absolute coverage error | ace | SQR vs QDT | 67020.5 | 0.4670 | No |
| | | SQR vs LQR | 53296.0 | $< 0.001$ | **Yes** |
| | | QDT vs LQR | 53257.0 | $< 0.001$ | **Yes** |
| Parsimony | pars | SQR vs QDT | 3187.0 | $< 0.001$ | **Yes** |
| | | SQR vs LQR | 39726.0 | $< 0.001$ | **Yes** |
| | | QDT vs LQR | 9425.0 | $< 0.001$ | **Yes** |

Table 9: Pairwise Wilcoxon Signed-Rank Test Results for $\tau = .9$.

| Metric | Abbr. | Models | $T$-statistic | p-value | Significant |
|---|---|---|---|---|---|
| Normalized quantile loss | nql | SQR vs QDT | 30982.0 | $< 0.001$ | **Yes** |
| | | SQR vs LQR | 8134.0 | $< 0.001$ | **Yes** |
| | | QDT vs LQR | 8059.0 | $< 0.001$ | **Yes** |
| Absolute coverage error | ace | SQR vs QDT | 39604.0 | $< 0.001$ | **Yes** |
| | | SQR vs LQR | 10842.5 | $< 0.001$ | **Yes** |
| | | QDT vs LQR | 14197.0 | $< 0.001$ | **Yes** |
| Parsimony | pars | SQR vs QDT | 11966.0 | $< 0.001$ | **Yes** |
| | | SQR vs LQR | 39098.0 | $< 0.001$ | **Yes** |
| | | QDT vs LQR | 30847.5 | $< 0.001$ | **Yes** |

- **H1**$_{3,50}$**:** SQR significantly outperforms other quantile regressors in terms of Model parsimony on the benchmark dataset.

For the 90th quantile:

- **H0**$_{1,90}$**:** SQR does not significantly outperform other quantile regressors in terms of Normalized Pinball Loss on the benchmark dataset.

- **H1**$_{1,90}$**:** SQR significantly outperforms other quantile regressors in terms of Normalized Pinball Loss on the benchmark dataset.

- **H0**$_{2,90}$**:** SQR does not significantly outperform other quantile regressors in terms of empirical coverage (Absolute Coverage Error) on the benchmark dataset.

- **H1**$_{2,90}$**:** SQR significantly outperforms other quantile regressors in terms of empirical coverage (Absolute Coverage Error) on the benchmark dataset.

- **H0**$_{3,90}$**:** SQR does not significantly outperform other quantile regressors in terms of Model parsimony on the benchmark dataset.

- **H1**$_{3,90}$**:** SQR significantly outperforms other quantile regressors in terms of Model parsimony on the benchmark dataset.

## D   Settings and hyperparameters

Table 10: PySRRegressor Parameters

| Parameter | Value | Parameter | Value |
|---|---|---|---|
| maxsize | 20 | fraction_replaced | 0.000364 |
| maxdepth | None | fraction_replaced_hof | 0.035 |
| niterations | 900 | migration | True |
| populations | 31 | hof_migration | True |
| population_size | 33 | topn | 12 |
| ncycles_per_iteration | 550 | denoise | False |
| model_selection | 'best' | select_k_features | None |
| dimensional_constraint_penalty | 1000.0 | max_evals | None |
| parsimony | 0.0 | timeout_in_seconds | None |
| constraints | None | early_stop_condition | None |
| nested_constraints | None | procs | cpu_count() |
| complexity_of_operators | None | multithreading | True |
| complexity_of_constants | 1 | cluster_manager | None |
| complexity_of_variables | 1 | heap_size_hint_in_bytes | None |
| warmup_maxsize_by | 0.0 | batching | False |
| use_frequency | True | batch_size | 50 |
| use_frequency_in_tournament | True | precision | 32 |
| adaptive_parsimony_scaling | 20.0 | fast_cycle | False |
| should_simplify | True | turbo | False |
| weight_add_node | 0.79 | bumper | False |
| weight_insert_node | 5.1 | enable_autodiff | False |
| weight_delete_node | 1.7 | random_state | None |
| weight_do_nothing | 0.21 | deterministic | False |
| weight_mutate_constant | 0.048 | warm_start | False |
| weight_mutate_operator | 0.47 | verbosity | 1 |
| weight_swap_operands | 0.1 | update_verbosity | None |
| weight_randomize | 0.00023 | print_precision | 5 |
| weight_simplify | 0.0020 | progress | True |
| weight_optimize | 0.0 | temp_equation_file | False |
| crossover_probability | 0.066 | tempdir | None |
| annealing | False | delete_tempfiles | True |
| alpha | 0.1 | update | False |
| perturbation_factor | 0.076 | tournament_selection_n | 10 |
| skip_mutation_failures | True | tournament_selection_p | 0.86 |
| optimizer_algorithm | "BFGS" | optimizer_nrestarts | 2 |
| optimize_probability | 0.14 | optimizer_iterations | 8 |
| should_optimize_constants | True | | |

Table 11: Hyperparameters optimized through Optuna for each model in the quantitative evaluation

| LGBM | |
|---|---|
| **Hyperparameter** | **Optimization Range** |
| num_leaves | 2 to 100 |
| learning_rate | 0.01 to 0.5 (log-uniform) |
| max_depth | 1 to 20 |
| min_child_samples | 5 to 100 |
| **QDT** | |
| min_samples_leaf | 1 to 50 |
| **LQR** | |
| max_iter | 1000 to 10000 |

# E   Additional Results

To supplement the aggregate performance metrics in Table 2, we here examine the empirical relationship between predictive performance, (parsimony), and dataset characteristics. Figure 6 visualizes how parsimony, calibration (in absolute coverage error) and predictive performance (in normalised quantile loss) scale as functions of the number of categorical features, sample size ($n$), and number of features ($d$). Table 12 shows the bin sizes for each type of plot.

The included models exhibit different scaling behaviors which may be explained by their respective inductive biases. LQR scales parsimony linearly with feature dimensionality ($d$), as the model generally assigns a coefficient to each available predictor. QDT's parsimony appears primarily driven by sample size ($n$); without strict pruning, these models continue to partition the feature space at a granular level when a large number of samples is available per tree, leading to high parsimony scores in large-$n$ regimes. SQR maintains a consistently low parsimony score that remains relatively invariant to changes in $n$ or $d$, empirically confirming its builtin preference for interpretable representations.

A relevant question now is how these inherent preferences for simple, interpretable models affect SQRs predictive performance in different data regimes. In high-dimensional regimes (large $d$ relative to $n$), the unconstrained growth of LQR may increase susceptibility to overfitting, potentially leading to higher Absolute Coverage Errors at extreme quantiles ($\tau = .9$), in contrast to SQR's robustness wrt calibration. As an example, on dataset ID 505 in Table 13 with $d = 124$, LQR yields a parsimony score of 124.00, whereas SQR maintains scores of 12.60 ($\tau = .5$) and 7.00 ($\tau = .9$). In such cases, SQR's constrained optimization approach appemain)rears to generate competitive normalized quantile loss and stable empirical coverage by isolating a restricted subset of informative predictors at lower parsimony.

Conversely, in large-sample regimes (large $n$ relative to $d$), the available data may support more complex, non-linear functional forms. Models with higher capacity, such as QDT and the LGBM baseline can model nuances better without overfitting due to the large sample sizes. For example, on dataset ID 344 ($n = 40,768$), QDT reaches an absolute coverage error of 0.01 at a parsimony of over twelve thousand. In contrast, SQR remains highly constrained (Parsimony = 11.60) at the cost of calibration with a coverage error of 0.1. These results suggest that SQR's restricted hypothesis space may lead to underfitting in large-$n$ regimes, resulting in a measurable increase in quantile Loss compared to more flexible, tree-based estimators.

These observations suggest that, while SQR provides a robust mechanism for regularization in high-dimensional or safety-critical tail estimations, its limited capacity may pose a predictive performance bottleneck when applied to large-scale datasets where higher-order interactions are present.

Table 12: Dataset distributions for bins in Figure 6.

| Variable | Size | Range | # Datasets |
|---|---|---|---|
| Categorical $d$ | 0 | $0 - 0$ | 388 |
| | S | $1.0 - 2.0$ | 36 |
| | L | $2.0 - 22.0$ | 32 |
| $d$ | S | $2.0 - 5.0$ | 140 |
| | M | $5.0 - 10.0$ | 148 |
| | L | $10.0 - 25.0$ | 100 |
| | XL | $25.0 - 124.0$ | 68 |
| $n$ | S | $47.0 - 240.0$ | 116 |
| | M | $240.0 - 500.0$ | 168 |
| | L | $500.0 - 1000.0$ | 92 |
| | XL | $1000.0 - 40768.0$ | 80 |

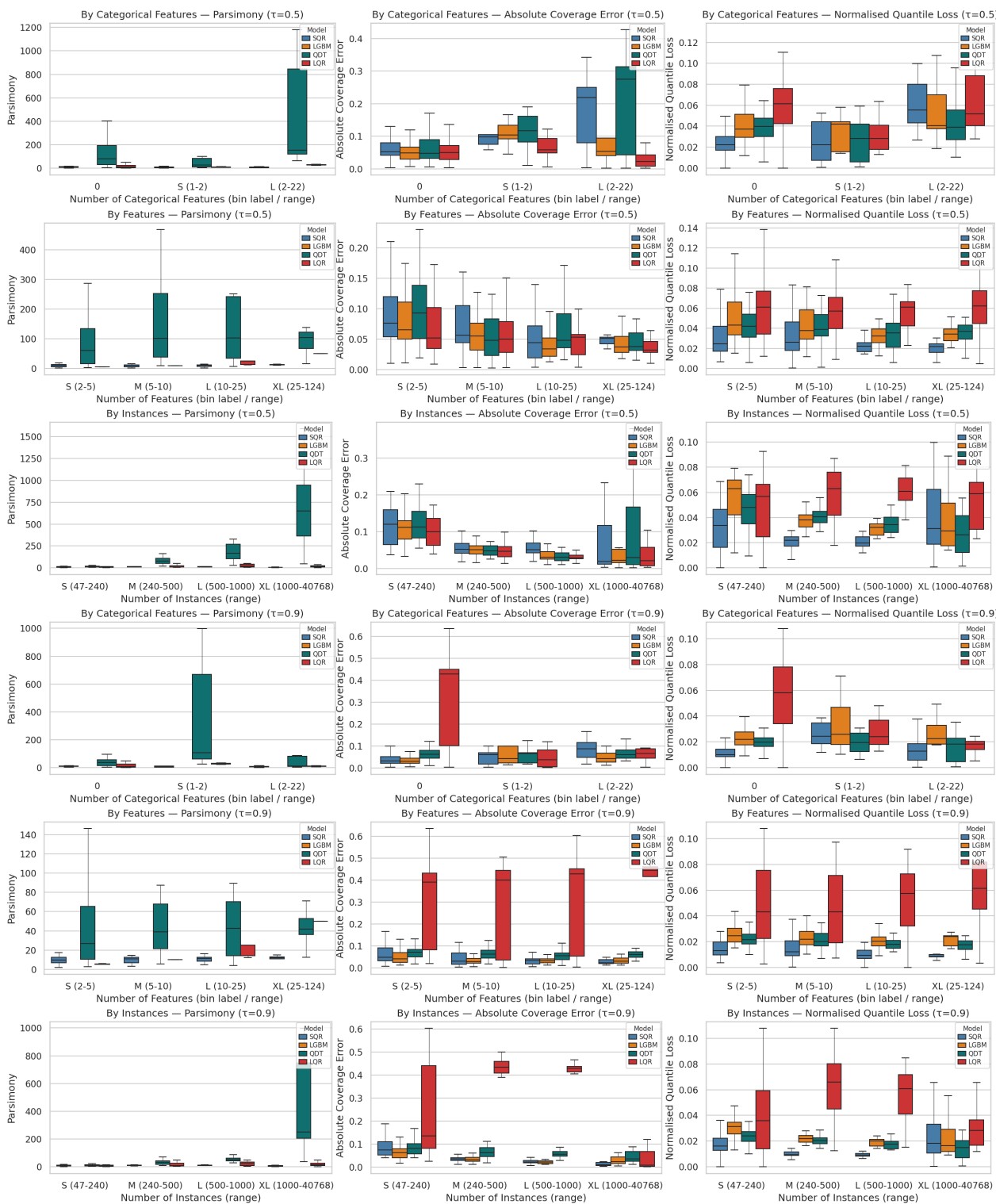

Figure 6: Detailed results grouped by number of categorical features, overall features, and instances.

Table 13: Extensive results overview

| τ | ID | n | d | Parsimony | | | Absolute Coverage Error | | | | Normalized Quantile Loss | | | | Time (ms) | | | |
|---|---|---|---|---|---|---|---|---|---|---|---|---|---|---|---|---|---|---|
| | | | | QDT | LQR | SQR | QDT | LGBM | LQR | SQR | QDT | LGBM | LQR | SQR | QDT | LGBM | LQR | SQR |
| .5 | 1027 | 488 | 4 | 127.00 | 34.00 | 8.60 | 0.28 | 0.05 | 0.02 | 0.30 | 0.03 | 0.04 | 0.03 | 0.04 | 0.05 | 0.18 | 1.55 | 417.37 |
| | 1028 | 1000 | 10 | 101.40 | 31.00 | 3.00 | 0.27 | 0.07 | 0.03 | 0.21 | 0.07 | 0.08 | 0.08 | 0.08 | 0.06 | 0.30 | 1.81 | 607.82 |
| | 1029 | 1000 | 4 | 165.80 | 20.00 | 3.00 | 0.30 | 0.05 | 0.03 | 0.23 | 0.05 | 0.07 | 0.06 | 0.08 | 0.08 | 0.35 | 1.24 | 644.40 |
| | 1030 | 1000 | 4 | 42.20 | 4.00 | 3.20 | 0.11 | 0.06 | 0.05 | 0.05 | 0.08 | 0.08 | 0.08 | 0.08 | 0.04 | 0.28 | 0.09 | 630.11 |
| | 1089 | 47 | 13 | 7.00 | 14.00 | 11.20 | 0.16 | 0.10 | 0.14 | 0.14 | 0.06 | 0.12 | 0.06 | 0.06 | 0.02 | 0.02 | 1.94 | 278.11 |
| | 1096 | 50 | 4 | 11.40 | 5.00 | 14.40 | 0.14 | 0.08 | 0.06 | 0.16 | 0.04 | 0.08 | 0.01 | 0.01 | 0.02 | 0.02 | 0.09 | 285.53 |
| | 1193 | 31104 | 9 | 787.80 | 23.00 | 3.20 | 0.01 | 0.01 | 0.00 | 0.00 | 0.04 | 0.04 | 0.04 | 0.05 | 1.74 | 39.87 | 9.50 | 46954.57 |
| | 1199 | 17496 | 9 | 467.80 | 12.00 | 2.00 | 0.01 | 0.03 | 0.01 | 0.15 | 0.06 | 0.06 | 0.06 | 0.06 | 1.40 | 72.87 | 4.09 | 11112.44 |
| | 192 | 52 | 2 | 7.00 | 2.00 | 6.40 | 0.09 | 0.17 | 0.17 | 0.06 | 0.05 | 0.07 | 0.06 | 0.04 | 0.02 | 0.02 | 0.06 | 280.82 |
| | 197 | 8192 | 21 | 580.20 | 21.00 | 4.00 | 0.11 | 0.02 | 0.06 | 0.02 | 0.01 | 0.02 | 0.06 | 0.02 | 1.27 | 1.07 | 48.82 | 5003.10 |
| | 201 | 15000 | 48 | 1180.20 | 48.00 | 10.40 | 0.43 | 0.18 | 0.01 | 0.23 | 0.01 | 0.03 | 0.13 | 0.07 | 0.82 | 2.07 | 110.24 | 6569.89 |
| | 210 | 108 | 5 | 27.00 | 9.00 | 5.00 | 0.08 | 0.09 | 0.09 | 0.09 | 0.03 | 0.03 | 0.02 | 0.02 | 0.02 | 0.07 | 0.17 | 309.27 |
| | 215 | 40768 | 10 | 1013.80 | 29.00 | 14.20 | 0.00 | 0.00 | 0.00 | 0.00 | 0.02 | 0.02 | 0.04 | 0.03 | 1.52 | 222.50 | 18.67 | 62236.88 |
| | 218 | 22784 | 8 | 722.20 | 8.00 | 5.00 | 0.02 | 0.02 | 0.01 | 0.04 | 0.02 | 0.02 | 0.02 | 0.02 | 2.10 | 55.59 | 23.86 | 21459.98 |
| | 225 | 8192 | 8 | 252.60 | 8.00 | 3.00 | 0.01 | 0.01 | 0.10 | 0.16 | 0.05 | 0.05 | 0.08 | 0.06 | 0.90 | 38.36 | 1.32 | 10376.78 |
| | 227 | 8192 | 12 | 744.20 | 12.00 | 5.60 | 0.09 | 0.02 | 0.06 | 0.01 | 0.01 | 0.02 | 0.06 | 0.02 | 0.99 | 1.00 | 6.54 | 4147.02 |
| | 228 | 55 | 2 | 8.20 | 2.00 | 7.40 | 0.14 | 0.08 | 0.14 | 0.17 | 0.05 | 0.07 | 0.14 | 0.04 | 0.02 | 0.04 | 0.11 | 285.09 |
| | 229 | 200 | 10 | 63.40 | 29.00 | 9.20 | 0.06 | 0.06 | 0.08 | 0.11 | 0.04 | 0.04 | 0.04 | 0.05 | 0.02 | 0.10 | 0.85 | 331.74 |
| | 230 | 209 | 6 | 64.60 | 6.00 | 5.00 | 0.08 | 0.12 | 0.15 | 0.11 | 0.02 | 0.02 | 0.02 | 0.01 | 0.02 | 0.10 | 0.87 | 332.86 |
| | 294 | 6435 | 36 | 402.20 | 36.00 | 5.40 | 0.42 | 0.05 | 0.03 | 0.02 | 0.03 | 0.05 | 0.11 | 0.09 | 0.95 | 0.87 | 6.28 | 2485.58 |
| | 344 | 40768 | 10 | 12764.20 | 14.00 | 11.60 | 0.01 | 0.06 | 0.01 | 0.10 | 0.00 | 0.01 | 0.03 | 0.00 | 3.48 | 19.61 | 3.56 | 54050.23 |
| | 4544 | 1059 | 117 | 43.40 | 117.00 | 4.20 | 0.03 | 0.03 | 0.03 | 0.03 | 0.03 | 0.03 | 0.02 | 0.03 | 0.63 | 0.52 | 58.03 | 682.21 |
| | 485 | 48 | 4 | 11.80 | 12.00 | 18.80 | 0.06 | 0.17 | 0.12 | 0.21 | 0.06 | 0.11 | 0.06 | 0.05 | 0.02 | 0.02 | 0.04 | 263.80 |
| | 503 | 6574 | 14 | 241.40 | 14.00 | 2.00 | 0.02 | 0.01 | 0.01 | 0.01 | 0.03 | 0.03 | 0.03 | 0.04 | 0.94 | 0.98 | 2.24 | 2449.37 |
| | 505 | 240 | 124 | 66.60 | 124.00 | 12.60 | 0.08 | 0.03 | 0.06 | 0.04 | 0.01 | 0.02 | 0.00 | 0.01 | 0.09 | 0.16 | 212.41 | 403.18 |
| | 519 | 380 | 2 | 35.40 | 5.00 | 2.00 | 0.11 | 0.04 | 0.05 | 0.06 | 0.04 | 0.04 | 0.04 | 0.04 | 0.03 | 0.14 | 0.10 | 416.16 |
| | 522 | 500 | 7 | 48.20 | 7.00 | 6.20 | 0.04 | 0.05 | 0.06 | 0.04 | 0.06 | 0.06 | 0.07 | 0.07 | 0.04 | 0.19 | 0.09 | 475.29 |
| | 523 | 100 | 2 | 3.00 | 6.00 | 3.00 | 0.37 | 0.13 | 0.25 | 0.21 | 0.03 | 0.04 | 0.03 | 0.03 | 0.01 | 0.05 | 0.03 | 286.35 |
| | 527 | 67 | 14 | 18.20 | 14.00 | 15.20 | 0.17 | 0.11 | 0.50 | 0.50 | 0.02 | 0.04 | 0.00 | 0.00 | 0.02 | 0.05 | 0.02 | 257.00 |
| | 529 | 3848 | 4 | 477.40 | 4.00 | 4.00 | 0.03 | 0.02 | 0.01 | 0.01 | 0.03 | 0.03 | 0.02 | 0.03 | 0.36 | 0.88 | 0.22 | 1670.74 |
| | 537 | 20640 | 8 | 1590.20 | 8.00 | 2.00 | 0.38 | 0.38 | 0.40 | 0.43 | 0.04 | 0.06 | 0.06 | 0.08 | NaN | NaN | NaN | NaN |
| | 542 | 60 | 15 | 7.80 | 15.00 | 8.80 | 0.17 | 0.07 | 0.13 | 0.17 | 0.06 | 0.06 | 0.07 | 0.07 | 0.02 | 0.05 | 0.50 | 286.53 |
| | 547 | 500 | 7 | 32.20 | 7.00 | 3.00 | 0.03 | 0.04 | 0.05 | 0.05 | 0.04 | 0.04 | 0.04 | 0.05 | 0.06 | 0.22 | 0.30 | 445.13 |
| | 556 | 475 | 3 | 101.80 | 11.00 | 9.00 | 0.16 | 0.10 | 0.06 | 0.10 | 0.01 | 0.02 | 0.02 | 0.01 | 0.05 | 0.18 | 0.12 | 382.88 |
| | 557 | 475 | 3 | 83.40 | 11.00 | 3.80 | 0.19 | 0.13 | 0.06 | 0.10 | 0.01 | 0.02 | 0.02 | 0.01 | 0.04 | 0.18 | 0.16 | 379.58 |
| | 560 | 252 | 14 | 79.80 | 14.00 | 10.80 | 0.04 | 0.05 | 0.10 | 0.07 | 0.01 | 0.01 | 0.03 | 0.00 | 0.04 | 0.12 | 0.91 | 325.69 |
| | 561 | 209 | 7 | 52.20 | 7.00 | 10.60 | 0.11 | 0.13 | 0.05 | 0.05 | 0.01 | 0.01 | 0.01 | 0.00 | 0.03 | 0.10 | 0.17 | 325.50 |
| | 562 | 8192 | 12 | 744.20 | 12.00 | 5.60 | 0.09 | 0.02 | 0.06 | 0.01 | 0.01 | 0.02 | 0.06 | 0.02 | 1.20 | 0.99 | 6.76 | 4128.40 |
| | 564 | 40768 | 10 | 5656.60 | 10.00 | 10.60 | 0.01 | 0.01 | 0.01 | 0.01 | 0.02 | 0.03 | 0.03 | 0.02 | 3.60 | 22.54 | 6.24 | 54467.68 |
| | 573 | 8192 | 21 | 580.20 | 21.00 | 4.00 | 0.11 | 0.02 | 0.06 | 0.02 | 0.01 | 0.02 | 0.06 | 0.02 | 1.55 | 1.14 | 51.64 | 4988.03 |
| | 574 | 22784 | 16 | 921.80 | 16.00 | 3.40 | 0.03 | 0.03 | 0.00 | 0.00 | 0.02 | 0.02 | 0.02 | 0.03 | 3.68 | 65.43 | 31.43 | 28777.25 |
| | 579 | 250 | 5 | 61.00 | 5.00 | 14.00 | 0.03 | 0.05 | 0.05 | 0.03 | 0.06 | 0.05 | 0.05 | 0.03 | 0.03 | 0.13 | 0.13 | 369.25 |
| | 581 | 500 | 25 | 104.60 | 25.00 | 10.80 | 0.03 | 0.02 | 0.05 | 0.07 | 0.04 | 0.03 | 0.06 | 0.02 | 0.09 | 0.22 | 0.95 | 587.65 |
| | 582 | 500 | 25 | 119.00 | 25.00 | 11.20 | 0.06 | 0.04 | 0.02 | 0.08 | 0.04 | 0.04 | 0.08 | 0.03 | 0.10 | 0.24 | 1.67 | 589.31 |
| | 583 | 1000 | 50 | 122.60 | 50.00 | 13.60 | 0.02 | 0.03 | 0.03 | 0.04 | 0.04 | 0.03 | 0.08 | 0.02 | 0.35 | 0.42 | 31.36 | 844.05 |
| | 584 | 500 | 25 | 140.60 | 25.00 | 13.00 | 0.04 | 0.04 | 0.01 | 0.04 | 0.04 | 0.03 | 0.07 | 0.01 | 0.09 | 0.24 | 0.78 | 598.48 |
| | 588 | 1000 | 100 | 138.60 | 100.00 | 13.00 | 0.03 | 0.04 | 0.01 | 0.05 | 0.03 | 0.02 | 0.06 | 0.01 | 0.63 | 0.49 | 43.78 | 833.32 |
| | 589 | 1000 | 25 | 194.20 | 25.00 | 10.40 | 0.02 | 0.03 | 0.02 | 0.05 | 0.04 | 0.03 | 0.07 | 0.03 | 0.19 | 0.39 | 1.56 | 954.59 |
| | 590 | 1000 | 50 | 104.60 | 50.00 | 13.80 | 0.04 | 0.04 | 0.04 | 0.05 | 0.05 | 0.04 | 0.04 | 0.02 | 0.34 | 0.44 | 79.81 | 742.22 |
| | 591 | 100 | 10 | 13.00 | 10.00 | 16.60 | 0.07 | 0.07 | 0.05 | 0.15 | 0.06 | 0.07 | 0.09 | 0.01 | 0.02 | 0.07 | 0.79 | 314.17 |
| | 592 | 1000 | 25 | 251.40 | 25.00 | 11.60 | 0.06 | 0.05 | 0.05 | 0.03 | 0.03 | 0.03 | 0.06 | 0.02 | 0.24 | 0.40 | 1.53 | 946.77 |
| | 593 | 1000 | 10 | 250.20 | 10.00 | 13.80 | 0.04 | 0.04 | 0.03 | 0.07 | 0.03 | 0.03 | 0.07 | 0.02 | 0.10 | 0.35 | 0.20 | 817.72 |
| | 594 | 100 | 5 | 17.00 | 5.00 | 12.80 | 0.16 | 0.13 | 0.17 | 0.12 | 0.06 | 0.07 | 0.09 | 0.04 | 0.02 | 0.07 | 0.16 | 314.67 |
| | 595 | 1000 | 10 | 101.00 | 10.00 | 14.00 | 0.02 | 0.03 | 0.05 | 0.02 | 0.04 | 0.03 | 0.04 | 0.02 | 0.12 | 0.37 | 0.27 | 680.13 |
| | 596 | 250 | 5 | 47.40 | 5.00 | 13.80 | 0.05 | 0.08 | 0.06 | 0.02 | 0.05 | 0.05 | 0.09 | 0.02 | 0.03 | 0.13 | 0.16 | 386.82 |
| | 597 | 500 | 5 | 107.80 | 5.00 | 14.40 | 0.06 | 0.06 | 0.04 | 0.04 | 0.04 | 0.04 | 0.08 | 0.02 | 0.06 | 0.21 | 0.16 | 504.96 |
| | 598 | 1000 | 25 | 108.60 | 25.00 | 14.20 | 0.04 | 0.02 | 0.03 | 0.04 | 0.05 | 0.04 | 0.04 | 0.02 | 0.26 | 0.40 | 1.08 | 885.33 |
| | 599 | 1000 | 5 | 329.40 | 5.00 | 11.40 | 0.04 | 0.03 | 0.03 | 0.04 | 0.03 | 0.03 | 0.08 | 0.02 | 0.10 | 0.33 | 0.28 | 768.03 |
| | 601 | 250 | 5 | 79.40 | 5.00 | 13.80 | 0.06 | 0.06 | 0.04 | 0.08 | 0.04 | 0.04 | 0.07 | 0.02 | 0.03 | 0.12 | 0.12 | 395.85 |
| | 602 | 250 | 10 | 63.00 | 10.00 | 9.80 | 0.06 | 0.08 | 0.09 | 0.04 | 0.04 | 0.04 | 0.07 | 0.02 | 0.03 | 0.13 | 0.16 | 378.80 |
| | 603 | 250 | 50 | 16.20 | 50.00 | 10.60 | 0.05 | 0.09 | 0.06 | 0.12 | 0.05 | 0.04 | 0.04 | 0.02 | 0.06 | 0.15 | 60.95 | 400.30 |
| | 604 | 500 | 10 | 122.60 | 10.00 | 11.80 | 0.04 | 0.06 | 0.06 | 0.05 | 0.03 | 0.03 | 0.06 | 0.02 | 0.06 | 0.22 | 0.38 | 499.31 |
| | 605 | 250 | 25 | 34.20 | 25.00 | 13.40 | 0.04 | 0.03 | 0.02 | 0.04 | 0.05 | 0.05 | 0.08 | 0.02 | 0.05 | 0.14 | 0.91 | 403.91 |
| | 606 | 1000 | 10 | 213.40 | 10.00 | 13.80 | 0.01 | 0.03 | 0.03 | 0.04 | 0.03 | 0.03 | 0.07 | 0.02 | 0.14 | 0.36 | 0.41 | 815.96 |
| | 607 | 1000 | 50 | 212.20 | 50.00 | 14.60 | 0.02 | 0.03 | 0.03 | 0.05 | 0.03 | 0.02 | 0.05 | 0.01 | 0.41 | 0.43 | 17.05 | 835.50 |
| | 608 | 1000 | 10 | 289.00 | 10.00 | 12.00 | 0.01 | 0.05 | 0.03 | 0.06 | 0.03 | 0.02 | 0.05 | 0.02 | 0.16 | 0.38 | 0.45 | 833.14 |
| | 609 | 1000 | 5 | 322.20 | 5.00 | 14.20 | 0.02 | 0.01 | 0.04 | 0.06 | 0.04 | 0.04 | 0.04 | 0.02 | 0.08 | 0.35 | 0.15 | 657.23 |
| | 611 | 100 | 5 | 21.40 | 5.00 | 13.20 | 0.12 | 0.05 | 0.04 | 0.10 | 0.05 | 0.06 | 0.08 | 0.02 | 0.02 | 0.06 | 0.15 | 320.75 |
| | 612 | 1000 | 5 | 286.60 | 5.00 | 15.00 | 0.02 | 0.03 | 0.03 | 0.09 | 0.03 | 0.03 | 0.07 | 0.01 | 0.14 | 0.36 | 0.26 | 765.92 |
| | 613 | 250 | 5 | 62.60 | 5.00 | 14.80 | 0.07 | 0.07 | 0.04 | 0.06 | 0.04 | 0.04 | 0.06 | 0.02 | 0.03 | 0.13 | 0.17 | 398.97 |
| | 615 | 250 | 10 | 80.60 | 10.00 | 10.80 | 0.06 | 0.06 | 0.07 | 0.05 | 0.04 | 0.04 | 0.06 | 0.02 | 0.03 | 0.13 | 0.17 | 378.79 |
| | 616 | 500 | 50 | 107.80 | 50.00 | 13.20 | 0.06 | 0.06 | 0.03 | 0.05 | 0.04 | 0.03 | 0.07 | 0.02 | 0.16 | 0.27 | 26.79 | 565.06 |
| | 617 | 500 | 5 | 143.80 | 5.00 | 13.20 | 0.04 | 0.05 | 0.02 | 0.07 | 0.03 | 0.03 | 0.06 | 0.01 | 0.07 | 0.22 | 0.20 | 505.50 |

| | | | | Parsimony | | | Absolute Coverage Error | | | | Quantile Loss | | | | Time (ms) | | | |
|---|---|---|---|---|---|---|---|---|---|---|---|---|---|---|---|---|---|---|
| $\tau$ | ID | $n$ | $d$ | QDT | LQR | SQR | QDT | LGBM | LQR | SQR | QDT | LGBM | LQR | SQR | QDT | LGBM | LQR | SQR |
| | 618 | 1000 | 50 | 105.80 | 50.00 | 13.40 | 0.02 | 0.03 | 0.03 | 0.05 | 0.03 | 0.03 | 0.06 | 0.02 | 0.41 | 0.42 | 59.76 | 855.20 |
| | 620 | 1000 | 25 | 102.20 | 25.00 | 12.80 | 0.03 | 0.03 | 0.02 | 0.02 | 0.04 | 0.03 | 0.07 | 0.02 | 0.21 | 0.40 | 1.23 | 952.61 |
| | 621 | 100 | 10 | 15.40 | 10.00 | 12.60 | 0.09 | 0.08 | 0.07 | 0.05 | 0.06 | 0.06 | 0.05 | 0.03 | 0.02 | 0.07 | 0.09 | 305.17 |
| | 622 | 1000 | 50 | 120.60 | 50.00 | 13.00 | 0.04 | 0.02 | 0.02 | 0.06 | 0.04 | 0.03 | 0.08 | 0.03 | 0.31 | 0.42 | 68.44 | 872.94 |
| | 623 | 1000 | 10 | 287.00 | 10.00 | 11.20 | 0.02 | 0.02 | 0.03 | 0.10 | 0.02 | 0.02 | 0.05 | 0.02 | 0.18 | 0.39 | 0.53 | 808.82 |
| | 624 | 100 | 5 | 15.80 | 5.00 | 13.80 | 0.09 | 0.14 | 0.13 | 0.12 | 0.05 | 0.06 | 0.04 | 0.03 | 0.02 | 0.06 | 0.27 | 311.88 |
| | 626 | 500 | 50 | 69.80 | 50.00 | 12.40 | 0.06 | 0.05 | 0.03 | 0.05 | 0.04 | 0.04 | 0.08 | 0.02 | 0.17 | 0.27 | 24.94 | 570.77 |
| | 627 | 500 | 10 | 125.00 | 10.00 | 14.20 | 0.05 | 0.06 | 0.03 | 0.05 | 0.04 | 0.03 | 0.07 | 0.02 | 0.06 | 0.22 | 0.17 | 488.32 |
| | 628 | 1000 | 5 | 320.60 | 5.00 | 13.00 | 0.04 | 0.03 | 0.03 | 0.08 | 0.03 | 0.02 | 0.06 | 0.02 | 0.10 | 0.35 | 0.19 | 766.35 |
| | 631 | 500 | 5 | 125.00 | 5.00 | 14.80 | 0.05 | 0.02 | 0.02 | 0.02 | 0.04 | 0.04 | 0.07 | 0.02 | 0.05 | 0.20 | 0.88 | 490.51 |
| | 633 | 500 | 25 | 41.80 | 25.00 | 12.40 | 0.03 | 0.02 | 0.06 | 0.04 | 0.05 | 0.04 | 0.04 | 0.03 | 0.10 | 0.23 | 1.91 | 536.16 |
| | 634 | 100 | 10 | 17.80 | 10.00 | 10.80 | 0.09 | 0.09 | 0.09 | 0.08 | 0.06 | 0.07 | 0.08 | 0.05 | 0.02 | 0.07 | 0.92 | 309.55 |
| | 635 | 250 | 10 | 31.40 | 10.00 | 11.60 | 0.05 | 0.04 | 0.05 | 0.06 | 0.05 | 0.05 | 0.05 | 0.03 | 0.04 | 0.13 | 0.25 | 359.96 |
| | 637 | 500 | 50 | 79.80 | 50.00 | 12.40 | 0.04 | 0.05 | 0.04 | 0.03 | 0.04 | 0.04 | 0.08 | 0.02 | 0.15 | 0.27 | 28.30 | 567.91 |
| | 641 | 500 | 10 | 161.80 | 10.00 | 14.80 | 0.03 | 0.03 | 0.06 | 0.07 | 0.04 | 0.04 | 0.08 | 0.01 | 0.08 | 0.23 | 0.48 | 495.00 |
| | 643 | 500 | 25 | 75.80 | 25.00 | 11.80 | 0.05 | 0.03 | 0.05 | 0.03 | 0.04 | 0.04 | 0.08 | 0.02 | 0.11 | 0.24 | 0.61 | 610.85 |
| | 644 | 250 | 25 | 54.20 | 25.00 | 12.40 | 0.04 | 0.05 | 0.03 | 0.05 | 0.04 | 0.04 | 0.08 | 0.02 | 0.04 | 0.13 | 0.60 | 417.86 |
| | 645 | 500 | 50 | 86.20 | 50.00 | 12.00 | 0.03 | 0.07 | 0.06 | 0.05 | 0.03 | 0.03 | 0.06 | 0.02 | 0.19 | 0.25 | 12.81 | 558.50 |
| | 646 | 500 | 10 | 161.40 | 10.00 | 13.20 | 0.06 | 0.05 | 0.03 | 0.05 | 0.03 | 0.03 | 0.06 | 0.01 | 0.09 | 0.24 | 0.66 | 494.46 |
| | 647 | 250 | 10 | 37.80 | 10.00 | 14.00 | 0.04 | 0.06 | 0.04 | 0.03 | 0.04 | 0.04 | 0.08 | 0.02 | 0.04 | 0.14 | 0.37 | 393.58 |
| | 648 | 250 | 50 | 53.40 | 50.00 | 13.40 | 0.05 | 0.04 | 0.08 | 0.05 | 0.05 | 0.04 | 0.09 | 0.02 | 0.08 | 0.15 | 8.67 | 424.57 |
| | 649 | 500 | 5 | 141.80 | 5.00 | 13.60 | 0.03 | 0.07 | 0.05 | 0.07 | 0.04 | 0.03 | 0.03 | 0.02 | 0.05 | 0.19 | 0.21 | 435.42 |
| | 650 | 500 | 50 | 46.60 | 50.00 | 13.80 | 0.04 | 0.02 | 0.05 | 0.04 | 0.05 | 0.04 | 0.04 | 0.02 | 0.13 | 0.25 | 58.40 | 513.67 |
| | 651 | 100 | 25 | 8.60 | 25.00 | 10.80 | 0.06 | 0.13 | 0.11 | 0.13 | 0.07 | 0.08 | 0.06 | 0.03 | 0.02 | 0.08 | 0.71 | 319.38 |
| | 653 | 250 | 25 | 23.80 | 25.00 | 12.20 | 0.06 | 0.05 | 0.08 | 0.10 | 0.05 | 0.05 | 0.04 | 0.02 | 0.04 | 0.14 | 0.95 | 389.50 |
| | 654 | 500 | 10 | 69.40 | 10.00 | 12.00 | 0.03 | 0.06 | 0.04 | 0.04 | 0.04 | 0.04 | 0.04 | 0.03 | 0.06 | 0.23 | 0.16 | 437.26 |
| | 656 | 100 | 5 | 25.00 | 5.00 | 11.40 | 0.11 | 0.17 | 0.10 | 0.12 | 0.06 | 0.07 | 0.09 | 0.03 | 0.02 | 0.07 | 0.38 | 316.60 |
| | 657 | 250 | 10 | 59.00 | 10.00 | 11.20 | 0.07 | 0.03 | 0.08 | 0.06 | 0.05 | 0.05 | 0.08 | 0.03 | 0.04 | 0.14 | 0.33 | 378.94 |
| | 658 | 250 | 25 | 43.80 | 25.00 | 11.80 | 0.04 | 0.04 | 0.04 | 0.06 | 0.04 | 0.04 | 0.06 | 0.02 | 0.04 | 0.15 | 1.29 | 411.38 |
| | 659 | 47 | 7 | 9.80 | 7.00 | 11.20 | 0.20 | 0.16 | 0.14 | 0.16 | 0.04 | 0.06 | 0.04 | 0.04 | 0.02 | 0.02 | 0.44 | 280.43 |
| | 663 | 120 | 2 | 18.60 | 11.00 | 14.80 | 0.12 | 0.09 | 0.06 | 0.07 | 0.04 | 0.04 | 0.01 | 0.01 | 0.02 | 0.07 | 0.11 | 314.80 |
| | 665 | 147 | 6 | 13.40 | 10.00 | 2.40 | 0.19 | 0.12 | 0.08 | 0.07 | 0.05 | 0.06 | 0.06 | 0.06 | 0.02 | 0.09 | 0.70 | 309.44 |
| | 666 | 508 | 10 | 29.00 | 10.00 | 2.40 | 0.06 | 0.05 | 0.04 | 0.04 | 0.03 | 0.03 | 0.03 | 0.03 | 0.05 | 0.20 | 0.49 | 469.00 |
| | 678 | 111 | 3 | 5.80 | 3.00 | 4.40 | 0.23 | 0.20 | 0.10 | 0.19 | 0.06 | 0.07 | 0.07 | 0.07 | 0.02 | 0.07 | 0.07 | 330.22 |
| | 687 | 62 | 5 | 10.60 | 5.00 | 6.80 | 0.10 | 0.07 | 0.13 | 0.05 | 0.06 | 0.08 | 0.06 | 0.07 | 0.02 | 0.05 | 0.09 | 291.70 |
| | 690 | 323 | 4 | 91.40 | 10.00 | 7.40 | 0.07 | 0.05 | 0.03 | 0.06 | 0.02 | 0.02 | 0.03 | 0.02 | 0.05 | 0.15 | 0.35 | 351.65 |
| | 695 | 235 | 12 | 11.40 | 12.00 | 6.80 | 0.06 | 0.07 | 0.05 | 0.06 | 0.03 | 0.03 | 0.02 | 0.02 | 0.03 | 0.11 | 0.29 | 364.61 |
| | 706 | 93 | 6 | 9.00 | 11.00 | 5.20 | 0.12 | 0.17 | 0.05 | 0.06 | 0.05 | 0.06 | 0.05 | 0.05 | 0.02 | 0.05 | 1.01 | 303.72 |
| | 712 | 222 | 2 | 11.80 | 2.00 | 3.40 | 0.11 | 0.12 | 0.09 | 0.12 | 0.05 | 0.05 | 0.09 | 0.05 | 0.02 | 0.10 | 0.09 | 385.31 |
| | banana | 5300 | 2 | 197.40 | 2.00 | 8.40 | 0.44 | 0.05 | 0.23 | 0.05 | 0.05 | 0.09 | 0.22 | 0.08 | 0.23 | 0.80 | 0.04 | 4244.90 |
| | titanic | 2099 | 8 | 143.00 | 31.00 | 5.20 | 0.35 | 0.18 | 0.15 | 0.34 | 0.10 | 0.11 | 0.11 | 0.10 | 0.17 | 0.57 | 0.83 | 859.30 |
| .9 | 1027 | 488 | 4 | 127.00 | 34.00 | 9.00 | 0.02 | 0.02 | 0.09 | 0.06 | 0.01 | 0.02 | 0.01 | 0.02 | 0.05 | 0.15 | 0.83 | 417.99 |
| | 1028 | 1000 | 10 | 65.40 | 31.00 | 3.00 | 0.07 | 0.10 | 0.04 | 0.09 | 0.03 | 0.04 | 0.03 | 0.03 | 0.06 | 0.05 | 2.02 | 632.44 |
| | 1029 | 1000 | 4 | 49.80 | 20.00 | 4.60 | 0.07 | 0.01 | 0.03 | 0.06 | 0.03 | 0.03 | 0.03 | 0.03 | 0.06 | 0.30 | 1.06 | 670.53 |
| | 1030 | 1000 | 4 | 34.60 | 4.00 | 4.20 | 0.04 | 0.03 | 0.05 | 0.04 | 0.03 | 0.04 | 0.04 | 0.04 | 0.05 | 0.28 | 0.21 | 601.44 |
| | 1089 | 47 | 13 | 7.00 | 14.00 | 9.40 | 0.14 | 0.07 | 0.23 | 0.15 | 0.02 | 0.05 | 0.06 | 0.02 | 0.02 | 0.03 | 0.61 | 278.51 |
| | 1096 | 50 | 4 | 8.60 | 5.00 | 13.20 | 0.18 | 0.04 | 0.16 | 0.06 | 0.02 | 0.05 | 0.00 | 0.00 | 0.02 | 0.03 | 0.06 | 272.76 |
| | 1193 | 31104 | 9 | 821.80 | 23.00 | 3.00 | 0.02 | 0.02 | 0.00 | 0.00 | 0.02 | 0.02 | 0.02 | 0.02 | 1.96 | 68.27 | 18.54 | 17203.20 |
| | 1199 | 17496 | 9 | 457.00 | 12.00 | 3.60 | 0.03 | 0.01 | 0.00 | 0.00 | 0.03 | 0.03 | 0.03 | 0.03 | 1.30 | 123.39 | 3.98 | 6687.39 |
| | 192 | 52 | 2 | 5.00 | 2.00 | 8.80 | 0.08 | 0.10 | 0.10 | 0.19 | 0.02 | 0.03 | 0.03 | 0.06 | 0.02 | 0.03 | 0.09 | 279.64 |
| | 197 | 8192 | 21 | 250.20 | 21.00 | 6.20 | 0.01 | 0.02 | 0.03 | 0.01 | 0.00 | 0.01 | 0.03 | 0.01 | 1.36 | 0.98 | 48.52 | 3736.67 |
| | 201 | 15000 | 48 | 619.80 | 48.00 | 11.40 | 0.07 | 0.10 | 0.01 | 0.02 | 0.01 | 0.07 | 0.05 | 0.04 | 1.28 | 0.19 | 104.02 | 5202.34 |
| | 210 | 108 | 5 | 11.00 | 9.00 | 7.40 | 0.08 | 0.07 | 0.08 | 0.05 | 0.02 | 0.02 | 0.01 | 0.01 | 0.02 | 0.08 | 0.19 | 300.22 |
| | 215 | 40768 | 10 | 997.80 | 29.00 | 12.80 | 0.02 | 0.05 | 0.00 | 0.01 | 0.01 | 0.01 | 0.02 | 0.01 | 2.01 | 453.39 | 15.57 | 35646.02 |
| | 218 | 22784 | 8 | 613.00 | 8.00 | 7.40 | 0.03 | 0.03 | 0.00 | 0.01 | 0.01 | 0.01 | 0.02 | 0.01 | 2.09 | 59.85 | 109.11 | 8624.05 |
| | 225 | 8192 | 8 | 210.20 | 8.00 | 4.20 | 0.02 | 0.01 | 0.50 | 0.01 | 0.02 | 0.02 | 0.09 | 0.02 | 0.70 | 66.04 | 1.36 | 6343.11 |
| | 227 | 8192 | 12 | 247.80 | 12.00 | 4.80 | 0.01 | 0.01 | 0.03 | 0.01 | 0.00 | 0.01 | 0.03 | 0.01 | 1.29 | 0.92 | 15.89 | 4341.78 |
| | 228 | 55 | 2 | 10.20 | 2.00 | 6.40 | 0.06 | 0.07 | 0.09 | 0.07 | 0.02 | 0.04 | 0.11 | 0.02 | 0.02 | 0.05 | 0.13 | 280.17 |
| | 229 | 200 | 10 | 27.40 | 29.00 | 13.00 | 0.12 | 0.04 | 0.08 | 0.06 | 0.02 | 0.02 | 0.02 | 0.02 | 0.03 | 0.10 | 0.55 | 326.27 |
| | 230 | 209 | 6 | 45.80 | 6.00 | 8.80 | 0.12 | 0.05 | 0.05 | 0.07 | 0.01 | 0.01 | 0.01 | 0.01 | 0.03 | 0.09 | 0.34 | 330.85 |
| | 294 | 6435 | 36 | 234.60 | 36.00 | 4.40 | 0.07 | 0.06 | 0.01 | 0.01 | 0.02 | 0.04 | 0.04 | 0.04 | 1.19 | 0.51 | 7.32 | 4016.42 |
| | 344 | 40768 | 10 | 11315.40 | 14.00 | 10.40 | 0.16 | 0.04 | 0.00 | 0.12 | 0.00 | 0.02 | 0.01 | 0.00 | 4.25 | 51.84 | 4.05 | 25496.77 |
| | 4544 | 1059 | 117 | 37.80 | 117.00 | 4.80 | 0.05 | 0.02 | 0.10 | 0.02 | 0.02 | 0.02 | 0.02 | 0.02 | 0.81 | 0.38 | 65.80 | 682.53 |
| | 485 | 48 | 4 | 10.60 | 12.00 | 17.20 | 0.13 | 0.13 | 0.09 | 0.11 | 0.04 | 0.06 | 0.04 | 0.04 | 0.02 | 0.08 | 0.81 | 261.71 |
| | 503 | 6574 | 14 | 204.60 | 14.00 | 2.00 | 0.04 | 0.02 | 0.01 | 0.00 | 0.02 | 0.02 | 0.01 | 0.02 | 0.75 | 0.81 | 4.18 | 2619.56 |
| | 505 | 240 | 124 | 41.80 | 124.00 | 7.00 | 0.07 | 0.04 | 0.15 | 0.05 | 0.01 | 0.03 | 0.00 | 0.00 | 0.10 | 0.12 | 207.59 | 402.95 |
| | 519 | 380 | 2 | 26.60 | 5.00 | 5.40 | 0.03 | 0.01 | 0.02 | 0.02 | 0.02 | 0.02 | 0.02 | 0.02 | 0.03 | 0.14 | 0.14 | 418.44 |
| | 522 | 500 | 7 | 18.20 | 7.00 | 5.80 | 0.05 | 0.02 | 0.02 | 0.02 | 0.03 | 0.03 | 0.03 | 0.03 | 0.05 | 0.18 | 0.28 | 486.79 |
| | 523 | 100 | 2 | 3.40 | 6.00 | 3.60 | 0.05 | 0.10 | 0.36 | 0.04 | 0.02 | 0.05 | 0.02 | 0.02 | 0.02 | 0.04 | 0.05 | 284.49 |
| | 527 | 67 | 14 | 15.40 | 14.00 | 16.20 | 0.17 | 0.08 | 0.60 | 0.10 | 0.02 | 0.03 | 0.00 | 0.00 | 0.02 | 0.06 | 0.02 | 251.01 |
| | 529 | 3848 | 4 | 177.00 | 4.00 | 5.40 | 0.05 | 0.01 | 0.39 | 0.01 | 0.01 | 0.02 | 0.02 | 0.01 | 0.29 | 0.68 | 0.21 | 1549.06 |
| | 537 | 20640 | 8 | 987.00 | 8.00 | 3.60 | 0.03 | 0.00 | 0.00 | 0.02 | 0.02 | 0.03 | 0.03 | 0.04 | 2.22 | 1.26 | 2.60 | 5396.63 |
| | 542 | 60 | 15 | 5.40 | 15.00 | 8.40 | 0.06 | 0.05 | 0.15 | 0.14 | 0.03 | 0.03 | 0.05 | 0.04 | 0.02 | 0.06 | 1.83 | 280.26 |
| | 547 | 500 | 7 | 31.80 | 7.00 | 5.80 | 0.04 | 0.02 | 0.05 | 0.03 | 0.02 | 0.02 | 0.02 | 0.02 | 0.06 | 0.19 | 0.30 | 451.30 |
| | 556 | 475 | 3 | 87.80 | 11.00 | 6.80 | 0.06 | 0.03 | 0.04 | 0.17 | 0.00 | 0.02 | 0.02 | 0.01 | 0.04 | 0.16 | 1.33 | 384.64 |

| $\tau$ | ID | $n$ | $d$ | Parsimony | | | Absolute Coverage Error | | | | Quantile Loss | | | | Time (ms) | | | |
|---|---|---|---|---|---|---|---|---|---|---|---|---|---|---|---|---|---|---|
| | | | | QDT | LQR | SQR | QDT | LGBM | LQR | SQR | QDT | LGBM | LQR | SQR | QDT | LGBM | LQR | SQR |
| | 557 | 475 | 3 | 83.00 | 11.00 | 7.00 | 0.04 | 0.03 | 0.04 | 0.14 | 0.00 | 0.02 | 0.02 | 0.01 | 0.06 | 0.17 | 1.18 | 377.32 |
| | 560 | 252 | 14 | 70.60 | 14.00 | 13.60 | 0.19 | 0.06 | 0.04 | 0.04 | 0.00 | 0.02 | 0.02 | 0.00 | 0.04 | 0.09 | 0.77 | 326.68 |
| | 561 | 209 | 7 | 58.60 | 7.00 | 10.00 | 0.09 | 0.06 | 0.03 | 0.04 | 0.01 | 0.01 | 0.01 | 0.00 | 0.03 | 0.08 | 0.94 | 328.90 |
| | 562 | 8192 | 12 | 247.80 | 12.00 | 4.80 | 0.01 | 0.01 | 0.03 | 0.01 | 0.00 | 0.01 | 0.03 | 0.01 | 1.18 | 0.90 | 16.45 | 4347.84 |
| | 564 | 40768 | 10 | 2332.60 | 10.00 | 11.40 | 0.06 | 0.03 | 0.01 | 0.00 | 0.01 | 0.01 | 0.01 | 0.01 | 4.29 | 207.23 | 10.34 | 25297.07 |
| | 574 | 22784 | 16 | 677.40 | 16.00 | 6.40 | 0.03 | 0.03 | 0.00 | 0.01 | 0.01 | 0.01 | 0.02 | 0.02 | 3.71 | 65.68 | 34.81 | 10718.79 |
| | 579 | 250 | 5 | 22.60 | 5.00 | 10.00 | 0.07 | 0.02 | 0.46 | 0.04 | 0.03 | 0.03 | 0.05 | 0.01 | 0.03 | 0.11 | 0.14 | 348.76 |
| | 581 | 500 | 25 | 57.00 | 25.00 | 14.20 | 0.11 | 0.03 | 0.46 | 0.04 | 0.02 | 0.02 | 0.07 | 0.01 | 0.09 | 0.18 | 1.46 | 577.45 |
| | 582 | 500 | 25 | 43.40 | 25.00 | 13.00 | 0.06 | 0.02 | 0.43 | 0.03 | 0.02 | 0.02 | 0.08 | 0.01 | 0.13 | 0.21 | 1.62 | 567.63 |
| | 583 | 1000 | 50 | 53.00 | 50.00 | 13.00 | 0.05 | 0.02 | 0.43 | 0.02 | 0.02 | 0.02 | 0.08 | 0.01 | 0.34 | 0.34 | 56.88 | 840.03 |
| | 584 | 500 | 25 | 42.60 | 25.00 | 11.20 | 0.11 | 0.04 | 0.43 | 0.02 | 0.02 | 0.02 | 0.07 | 0.01 | 0.11 | 0.19 | 2.91 | 614.10 |
| | 586 | 1000 | 25 | 89.40 | 25.00 | 14.00 | 0.09 | 0.02 | 0.42 | 0.01 | 0.01 | 0.02 | 0.06 | 0.01 | 0.19 | 0.29 | 0.96 | 938.14 |
| | 588 | 1000 | 100 | 55.80 | 100.00 | 12.20 | 0.06 | 0.01 | 0.45 | 0.02 | 0.01 | 0.02 | 0.07 | 0.01 | 0.69 | 0.36 | 92.25 | 807.68 |
| | 589 | 1000 | 25 | 61.80 | 25.00 | 12.00 | 0.06 | 0.02 | 0.43 | 0.02 | 0.02 | 0.02 | 0.07 | 0.01 | 0.18 | 0.32 | 1.08 | 1058.47 |
| | 590 | 1000 | 50 | 36.20 | 50.00 | 11.00 | 0.05 | 0.02 | 0.47 | 0.02 | 0.02 | 0.02 | 0.04 | 0.01 | 0.33 | 0.34 | 82.32 | 663.92 |
| | 591 | 100 | 10 | 8.60 | 10.00 | 12.00 | 0.08 | 0.09 | 0.40 | 0.07 | 0.03 | 0.03 | 0.10 | 0.01 | 0.03 | 0.08 | 0.41 | 312.11 |
| | 592 | 1000 | 25 | 52.20 | 25.00 | 11.20 | 0.06 | 0.03 | 0.44 | 0.02 | 0.02 | 0.02 | 0.06 | 0.01 | 0.22 | 0.29 | 1.74 | 928.71 |
| | 593 | 1000 | 10 | 61.00 | 10.00 | 13.80 | 0.07 | 0.01 | 0.44 | 0.02 | 0.02 | 0.02 | 0.07 | 0.01 | 0.15 | 0.30 | 0.35 | 827.27 |
| | 594 | 100 | 5 | 9.00 | 5.00 | 13.80 | 0.04 | 0.04 | 0.49 | 0.04 | 0.02 | 0.03 | 0.11 | 0.01 | 0.02 | 0.08 | 0.28 | 319.98 |
| | 595 | 1000 | 10 | 38.60 | 10.00 | 10.00 | 0.06 | 0.03 | 0.41 | 0.01 | 0.02 | 0.02 | 0.04 | 0.01 | 0.15 | 0.31 | 0.39 | 624.37 |
| | 596 | 250 | 5 | 15.40 | 5.00 | 12.40 | 0.09 | 0.05 | 0.40 | 0.07 | 0.03 | 0.03 | 0.09 | 0.01 | 0.03 | 0.13 | 0.10 | 404.04 |
| | 597 | 500 | 5 | 32.60 | 5.00 | 12.20 | 0.05 | 0.02 | 0.42 | 0.03 | 0.02 | 0.02 | 0.08 | 0.01 | 0.05 | 0.19 | 0.25 | 521.49 |
| | 598 | 1000 | 25 | 45.80 | 25.00 | 12.60 | 0.04 | 0.01 | 0.43 | 0.03 | 0.02 | 0.02 | 0.05 | 0.01 | 0.20 | 0.31 | 0.79 | 761.18 |
| | 599 | 1000 | 5 | 77.80 | 5.00 | 13.80 | 0.08 | 0.02 | 0.44 | 0.02 | 0.02 | 0.02 | 0.08 | 0.01 | 0.11 | 0.30 | 0.20 | 789.88 |
| | 601 | 250 | 5 | 16.60 | 5.00 | 13.40 | 0.04 | 0.04 | 0.44 | 0.05 | 0.02 | 0.03 | 0.08 | 0.01 | 0.03 | 0.12 | 0.13 | 398.24 |
| | 602 | 250 | 10 | 39.00 | 10.00 | 9.60 | 0.15 | 0.04 | 0.50 | 0.03 | 0.02 | 0.02 | 0.07 | 0.01 | 0.04 | 0.13 | 0.39 | 368.25 |
| | 603 | 250 | 50 | 12.60 | 50.00 | 11.40 | 0.04 | 0.04 | 0.46 | 0.04 | 0.02 | 0.02 | 0.05 | 0.01 | 0.05 | 0.13 | 45.19 | 367.35 |
| | 604 | 500 | 10 | 67.80 | 10.00 | 12.40 | 0.10 | 0.04 | 0.46 | 0.01 | 0.02 | 0.02 | 0.06 | 0.01 | 0.07 | 0.19 | 0.80 | 459.87 |
| | 605 | 250 | 25 | 10.60 | 25.00 | 12.20 | 0.03 | 0.03 | 0.43 | 0.05 | 0.02 | 0.03 | 0.09 | 0.01 | 0.04 | 0.13 | 0.97 | 432.07 |
| | 606 | 1000 | 10 | 43.40 | 10.00 | 10.00 | 0.05 | 0.03 | 0.43 | 0.02 | 0.02 | 0.02 | 0.07 | 0.01 | 0.16 | 0.32 | 0.73 | 875.80 |
| | 607 | 1000 | 50 | 71.40 | 50.00 | 13.40 | 0.08 | 0.02 | 0.42 | 0.02 | 0.01 | 0.01 | 0.06 | 0.01 | 0.39 | 0.33 | 27.36 | 829.80 |
| | 608 | 1000 | 10 | 53.00 | 10.00 | 12.60 | 0.05 | 0.02 | 0.44 | 0.01 | 0.01 | 0.01 | 0.06 | 0.01 | 0.13 | 0.27 | 0.71 | 767.77 |
| | 609 | 1000 | 5 | 55.80 | 5.00 | 9.60 | 0.05 | 0.03 | 0.40 | 0.03 | 0.02 | 0.02 | 0.04 | 0.01 | 0.12 | 0.30 | 0.34 | 589.04 |
| | 611 | 100 | 5 | 11.40 | 5.00 | 12.00 | 0.08 | 0.13 | 0.44 | 0.08 | 0.03 | 0.03 | 0.09 | 0.01 | 0.02 | 0.07 | 0.18 | 307.26 |
| | 612 | 1000 | 5 | 96.80 | 5.00 | 13.60 | 0.08 | 0.01 | 0.43 | 0.02 | 0.02 | 0.02 | 0.07 | 0.01 | 0.11 | 0.28 | 0.36 | 786.96 |
| | 613 | 250 | 5 | 34.20 | 5.00 | 11.80 | 0.09 | 0.04 | 0.42 | 0.03 | 0.02 | 0.02 | 0.07 | 0.01 | 0.03 | 0.13 | 0.13 | 387.20 |
| | 615 | 250 | 10 | 23.00 | 10.00 | 12.60 | 0.07 | 0.06 | 0.44 | 0.06 | 0.02 | 0.02 | 0.06 | 0.01 | 0.03 | 0.12 | 0.36 | 352.36 |
| | 616 | 500 | 50 | 37.00 | 50.00 | 12.40 | 0.08 | 0.03 | 0.44 | 0.03 | 0.02 | 0.02 | 0.08 | 0.01 | 0.17 | 0.21 | 51.08 | 551.43 |
| | 617 | 500 | 5 | 71.40 | 5.00 | 9.60 | 0.09 | 0.02 | 0.41 | 0.04 | 0.01 | 0.02 | 0.06 | 0.01 | 0.05 | 0.17 | 0.41 | 480.17 |
| | 618 | 1000 | 50 | 46.20 | 50.00 | 12.20 | 0.03 | 0.02 | 0.44 | 0.03 | 0.01 | 0.02 | 0.07 | 0.01 | 0.45 | 0.31 | 46.20 | 792.32 |
| | 620 | 1000 | 25 | 28.20 | 25.00 | 12.60 | 0.03 | 0.03 | 0.43 | 0.02 | 0.02 | 0.02 | 0.07 | 0.01 | 0.20 | 0.32 | 1.44 | 955.78 |
| | 621 | 100 | 10 | 7.40 | 10.00 | 11.60 | 0.05 | 0.02 | 0.46 | 0.08 | 0.03 | 0.03 | 0.05 | 0.01 | 0.02 | 0.08 | 0.21 | 287.23 |
| | 622 | 1000 | 50 | 43.40 | 50.00 | 13.40 | 0.03 | 0.03 | 0.43 | 0.02 | 0.02 | 0.02 | 0.08 | 0.01 | 0.45 | 0.37 | 37.83 | 931.59 |
| | 623 | 1000 | 10 | 57.40 | 10.00 | 13.20 | 0.05 | 0.02 | 0.42 | 0.02 | 0.01 | 0.01 | 0.05 | 0.01 | 0.16 | 0.29 | 0.64 | 783.34 |
| | 624 | 100 | 5 | 8.60 | 5.00 | 10.80 | 0.07 | 0.07 | 0.48 | 0.13 | 0.03 | 0.03 | 0.05 | 0.02 | 0.02 | 0.08 | 0.27 | 289.30 |
| | 626 | 500 | 50 | 40.20 | 50.00 | 14.60 | 0.09 | 0.05 | 0.44 | 0.04 | 0.02 | 0.02 | 0.09 | 0.01 | 0.17 | 0.22 | 18.70 | 616.20 |
| | 627 | 500 | 10 | 33.00 | 10.00 | 13.40 | 0.06 | 0.03 | 0.43 | 0.03 | 0.02 | 0.02 | 0.07 | 0.01 | 0.06 | 0.20 | 0.33 | 513.07 |
| | 628 | 1000 | 5 | 127.00 | 5.00 | 9.40 | 0.09 | 0.01 | 0.43 | 0.02 | 0.01 | 0.02 | 0.06 | 0.01 | 0.13 | 0.27 | 0.33 | 743.16 |
| | 631 | 500 | 5 | 28.60 | 5.00 | 14.60 | 0.05 | 0.03 | 0.39 | 0.03 | 0.02 | 0.02 | 0.07 | 0.01 | 0.05 | 0.17 | 0.17 | 501.60 |
| | 633 | 500 | 25 | 24.60 | 25.00 | 11.20 | 0.06 | 0.03 | 0.47 | 0.03 | 0.02 | 0.02 | 0.04 | 0.01 | 0.09 | 0.20 | 1.56 | 515.20 |
| | 634 | 100 | 10 | 15.40 | 10.00 | 11.40 | 0.07 | 0.02 | 0.45 | 0.11 | 0.03 | 0.03 | 0.09 | 0.02 | 0.02 | 0.07 | 0.27 | 322.81 |
| | 635 | 250 | 10 | 15.80 | 10.00 | 10.80 | 0.04 | 0.03 | 0.48 | 0.06 | 0.02 | 0.02 | 0.05 | 0.01 | 0.04 | 0.12 | 0.29 | 339.19 |
| | 637 | 500 | 50 | 26.60 | 50.00 | 12.20 | 0.05 | 0.04 | 0.45 | 0.04 | 0.02 | 0.02 | 0.09 | 0.01 | 0.14 | 0.24 | 54.62 | 576.31 |
| | 641 | 500 | 10 | 51.00 | 10.00 | 14.00 | 0.07 | 0.03 | 0.45 | 0.03 | 0.02 | 0.02 | 0.08 | 0.01 | 0.06 | 0.19 | 0.55 | 473.30 |
| | 643 | 500 | 25 | 30.20 | 25.00 | 13.40 | 0.04 | 0.04 | 0.45 | 0.04 | 0.02 | 0.02 | 0.08 | 0.01 | 0.11 | 0.20 | 1.60 | 647.74 |
| | 644 | 250 | 25 | 14.20 | 25.00 | 10.20 | 0.04 | 0.04 | 0.42 | 0.04 | 0.02 | 0.02 | 0.08 | 0.01 | 0.05 | 0.13 | 1.41 | 419.07 |
| | 645 | 500 | 50 | 41.80 | 50.00 | 15.00 | 0.09 | 0.03 | 0.48 | 0.03 | 0.02 | 0.02 | 0.06 | 0.01 | 0.14 | 0.21 | 10.35 | 529.98 |
| | 646 | 500 | 10 | 38.60 | 10.00 | 12.80 | 0.08 | 0.03 | 0.42 | 0.02 | 0.02 | 0.02 | 0.06 | 0.01 | 0.06 | 0.17 | 0.29 | 461.38 |
| | 647 | 250 | 10 | 21.40 | 10.00 | 14.00 | 0.06 | 0.02 | 0.44 | 0.03 | 0.02 | 0.02 | 0.08 | 0.01 | 0.04 | 0.13 | 0.84 | 381.77 |
| | 648 | 250 | 50 | 13.40 | 50.00 | 12.00 | 0.06 | 0.06 | 0.50 | 0.08 | 0.02 | 0.03 | 0.11 | 0.01 | 0.05 | 0.14 | 49.37 | 424.64 |
| | 649 | 500 | 5 | 43.00 | 5.00 | 10.40 | 0.06 | 0.04 | 0.43 | 0.04 | 0.02 | 0.02 | 0.03 | 0.01 | 0.06 | 0.17 | 0.47 | 405.27 |
| | 650 | 500 | 50 | 21.00 | 50.00 | 11.00 | 0.05 | 0.02 | 0.46 | 0.02 | 0.02 | 0.02 | 0.05 | 0.01 | 0.15 | 0.22 | 17.77 | 466.25 |
| | 651 | 100 | 25 | 3.80 | 25.00 | 10.00 | 0.04 | 0.05 | 0.51 | 0.07 | 0.03 | 0.03 | 0.09 | 0.01 | 0.02 | 0.08 | 1.54 | 297.21 |
| | 653 | 250 | 25 | 13.40 | 25.00 | 11.40 | 0.05 | 0.02 | 0.48 | 0.04 | 0.02 | 0.03 | 0.05 | 0.01 | 0.05 | 0.13 | 1.21 | 362.26 |
| | 654 | 500 | 10 | 17.80 | 10.00 | 11.00 | 0.04 | 0.03 | 0.43 | 0.03 | 0.02 | 0.02 | 0.04 | 0.01 | 0.05 | 0.18 | 0.25 | 396.92 |
| | 656 | 100 | 5 | 13.00 | 5.00 | 11.60 | 0.08 | 0.04 | 0.47 | 0.10 | 0.02 | 0.03 | 0.09 | 0.01 | 0.02 | 0.08 | 0.26 | 325.08 |
| | 657 | 250 | 10 | 32.20 | 10.00 | 14.60 | 0.08 | 0.04 | 0.46 | 0.04 | 0.02 | 0.03 | 0.09 | 0.01 | 0.03 | 0.13 | 0.25 | 392.79 |
| | 658 | 250 | 25 | 10.20 | 25.00 | 12.60 | 0.06 | 0.04 | 0.45 | 0.04 | 0.02 | 0.02 | 0.07 | 0.01 | 0.04 | 0.14 | 1.01 | 416.06 |
| | 659 | 47 | 7 | 7.00 | 7.00 | 12.40 | 0.14 | 0.06 | 0.09 | 0.11 | 0.03 | 0.03 | 0.02 | 0.02 | 0.02 | 0.04 | 0.50 | 270.21 |
| | 663 | 120 | 2 | 9.80 | 11.00 | 14.20 | 0.08 | 0.06 | 0.07 | 0.05 | 0.02 | 0.02 | 0.01 | 0.00 | 0.02 | 0.08 | 0.39 | 306.99 |
| | 665 | 147 | 6 | 11.40 | 10.00 | 6.80 | 0.05 | 0.04 | 0.04 | 0.07 | 0.03 | 0.04 | 0.04 | 0.04 | 0.02 | 0.09 | 0.31 | 306.20 |
| | 666 | 508 | 10 | 29.00 | 10.00 | 3.00 | 0.06 | 0.02 | 0.04 | 0.04 | 0.01 | 0.01 | 0.02 | 0.01 | 0.07 | 0.17 | 0.25 | 481.50 |
| | 678 | 111 | 3 | 2.60 | 3.00 | 3.20 | 0.09 | 0.11 | 0.13 | 0.12 | 0.03 | 0.04 | 0.04 | 0.04 | 0.02 | 0.09 | 0.08 | 324.77 |
| | 687 | 62 | 5 | 5.80 | 5.00 | 7.00 | 0.10 | 0.06 | 0.04 | 0.17 | 0.03 | 0.03 | 0.03 | 0.04 | 0.02 | 0.06 | 0.15 | 282.65 |
| | 690 | 323 | 4 | 59.80 | 10.00 | 7.00 | 0.07 | 0.03 | 0.04 | 0.03 | 0.01 | 0.02 | 0.01 | 0.01 | 0.04 | 0.12 | 0.45 | 357.64 |
| | 695 | 235 | 12 | 14.20 | 12.00 | 2.00 | 0.07 | 0.03 | 0.05 | 0.04 | 0.01 | 0.02 | 0.01 | 0.01 | 0.03 | 0.11 | 0.19 | 359.22 |

| $\tau$ ID | $n$ | $d$ | Parsimony | | | Absolute Coverage Error | | | | Quantile Loss | | | | Time (ms) | | | |
|---|---|---|---|---|---|---|---|---|---|---|---|---|---|---|---|---|---|
| | | | QDT | LQR | SQR | QDT | LGBM | LQR | SQR | QDT | LGBM | LQR | SQR | QDT | LGBM | LQR | SQR |
| 706 | 93 | 6 | 5.40 | 11.00 | 8.60 | 0.04 | 0.02 | 0.09 | 0.09 | 0.03 | 0.03 | 0.02 | 0.02 | 0.02 | 0.07 | 0.85 | 300.67 |
| 712 | 222 | 2 | 24.60 | 2.00 | 3.20 | 0.04 | 0.08 | 0.11 | 0.05 | 0.02 | 0.02 | 0.06 | 0.02 | 0.03 | 0.12 | 0.06 | 376.74 |
| banana | 5300 | 2 | 146.60 | 2.00 | 12.60 | 0.09 | 0.10 | 0.64 | 0.35 | 0.03 | 0.06 | 0.40 | 0.04 | 0.20 | 0.13 | 0.05 | 2141.00 |
| titanic | 2099 | 8 | 87.40 | 31.00 | 3.20 | 0.07 | 0.10 | 0.12 | 0.10 | 0.07 | 0.07 | 0.07 | 0.07 | 0.13 | 0.11 | 1.01 | 903.90 |

## F   Reproducibility

All experiments were conducted on a local machine with the following specifications:

- **Operating System:** Microsoft Windows 10 Enterprise

- **Programming Language:** Python (with PySR using a Julia backbone)

- **Libraries and Frameworks:**

    – PySR
    – lightgbm
    – scikit-learn
    – LogisticRegression (from sklearn.linear_model)
    – KFold (from sklearn.model_selection)
    – NumPy
    – Matplotlib
    – Seaborn
    – pandas
    – pmlb (Python Package for accessing PMLB datasets)
    – optuna
    – datetime (from Python standard library)

