# OpenReview forum: "Symbolic Quantile Regression for the Interpretable Prediction of Conditional Quantiles"
_TMLR — Accepted by TMLR_

### Review · Reviewer_48Jj · 2025-10-03

**Summary Of Contributions:**

The paper presents the use of symbolic regression for modelling quantiles. The paper combines two existing techniques: symbolic regression as implemented and provided by the existing package PySR nad pinball loss, which is well established loss for this type of the problem.

While the scientific contribution is weak part of the piper, the experimental evaluation is thorough and nice.

**Audience:**

Yes

**Audience Explanation:**

I think this is interesting approach and showing that symbolic regression can actually solve real-world problem is interesting.
Would be nice to try it beyond toy examples.

**Broader Impact Concerns:**

I do not view any ethical concerns.

**Claims And Evidence:**

No

**Claims Explanation:**

I am not that all convinces that the explanation as provided by the symbolic regression is all that great. Looking at Equation (11), it is nice and compact, yet I am not sure I can understand it. Specifically in Equation (11) I do not understand why there is a quadratic term with ASF, but it does not depend on the traveled distance. This suggest to me to be an artifact of fitting the data and indicates that the method will not extrapolate well beyond the support of the data (which is indeed paramount to most ML methods).

Possibly a good experiment might be to test extrapolation, or optimize the structure of the expressions such that after finetuning of coefficients by LBFS it would generalize well.

**Requested Changes:**

I would like to see a generalization test, where the algorithms are tested on data outside the training region.

---

> ### Author Response · Authors · 2026-02-26
> **Response to 48Jj**
>
> We are thankful to reviewer 48Jj’s critical comments, constructive concerns and appreciation of this work and its empirical evaluation on 122 real-world datasets and a novel airline fuel prediction use case as relevant to the TMLR audience. We agree that a better understanding of how SQR extrapolates beyond the support of the training data would be an interesting experiment, have executed such an experiment (Table 4 and Section 5.3) and find that SQR maintains its position as overall significantly best performing model at a relative moderate predictive performance penalty, for details see below.
>
> We now turn to the confusion about the quadratic ASF term in Equation 11 (Equation 10 in the revision) and its additive rather than multiplicative relation with traveled distance GCD. We hypothesize that this relation can be explained by aerodynamic principles stating that drag and hence the fuel consumption rate scale non-linearly with velocity in combination with the extra fuel used during high-thrust speeding phases (often during climb or specific catch-up segments). It therefore effectively acts as a distinct operational cost relative to the standard cruise efficiency. In consultation with domain experts, we further believe this additive rather than multiplicative relation can be partially explained from two airline practices to make up for time. In the first, a shallower climbing angle is selected to gain more forward motion compared to the relative steep climbing angle which is generally favoured as this results in reaching low-drag atmospheric zones earlier. In the second, a faster yet more fuel intensive descent is favoured over a slow fuel-efficient downglide during descent. To our understanding, the fuel usage during these two phases can be significant and is independent from the travel distance.
>
> To assess extrapolation beyond the training set support, we propose the following additional large-scale experiment. We first select LGBM on the basis of the overall best predictive performance and train it on each dataset. We calculate the feature importances using Shapley values for the best predictive model on each dataset. We then use the most predictive feature to split the data into a train and test split. Specifically the split is made at the 90th quantile of the most informative feature whereby all values for this feature below the 90th quantile make up the train split and all values higher form the out-of-distribution test split. To limit the computational burden, we run the above experimental setup on a random sample of 10k data points and assess the normalized quantile loss and absolute coverage error for all models. The results indicate that all models incur a performance penalty but that our approach remains the best-performing transparent model at comparative moderate cost. This can be explained by the enforced simplicity of the model acting as a regularizer to avoid overfitting.

---

### Review · Reviewer_Hx61 · 2025-10-27

**Summary Of Contributions:**

This work demonstrates the use of PySR, a symbolic regression (SR) optimization framework developed by Cranmer (2023), in interpretable quantile regression. In this context, interpretability is evaluated based on the parsimony score of the generated prediction function. The authors call their method symbolic quantile regression (SQR) and show that it mostly surpasses the performance of a linear quantile regressor (LQR) and a quantile decision tree (QDT), both of which have interpretable prediction functions, as well as a black-box LGBM quantile regressor when predicting medians and 90th quantiles. These results were evaluated on the SRBench dataset and scored with normalized quantile loss, absolute coverage error, and parsimony. Although SQR's runtime is around 100 times longer than that of the other methods, the authors address this limitation by training a model with only 10,000 samples from the training sets. This approach achieves similar performance while reducing the longer runtime by a factor of 10. Further optimization is discussed.

Strengths and Weaknesses:

- The paper presents an interesting and novel implementation of symbolic regression for quantile regression and is overall well-written and easy to follow.
- The experimental case study underscores the relevance of interpretable quantile regression and the proposed method.
- The experimental details seem to be fully documented but not adequately summarized and addressed in the main paper. The Optimization (3.3) subsection lacks a (brief) technical overview of the SQR model's implementation using PySR.
- Calling the main contribution an extension of SR appears to be a bit misleading, as it technically is an application of SR for QR and does not generalize the methods of SR.
- SQR is mainly compared to LQR (2010) and DQT (2011), and the related work section lacks a discussion of other work on (interpretable) quantile regression, if there is any.
- The experiments focus on 50% and 90% quantile functions without a discussion of why this is sufficient or generalizes to other quantiles.

**Audience:**

Yes

**Audience Explanation:**

Quantile regression is a very interesting and versatile use case for symbolic regression. It produces more interpretable and verifiable predictions from machine learning models, which makes them easier to adopt safely in several domains.

**Broader Impact Concerns:**

No concerns

**Claims And Evidence:**

No

**Claims Explanation:**

The description and evaluation of the SQR method for predicting median and 90th quantile functions are promising. However, the paper lacks a discussion of how these results support the conclusion that other quantiles can be modeled with similar performance. For example, while the differences in absolute coverage error between the median and the 90th quantile are mentioned, they are not discussed or explained in relation to other quantiles.

**Requested Changes:**

Critical Changes:

1.  The Optimization (3.3) subsection should have at least a brief technical overview of the SQR model's implementation using PySR and might be better called "Implementation".
2.  A short discussion of related work on quantile regression should be added to section 2.
3.  The Evaluation section should mention or even assess the potential to generalize to other quantiles than the 50th and 90th.
4.  The limitation of higher runtime should be addressed in the discussion.

Minor Changes:

1.  The comments on the different baseline models (p.6) should be backed by a reference.
2.  Please add your references to the different parts of your appendix within the text, not just at the beginning of the section.
3.  The statistical test evaluation summary can be misleading because its typography is similar to that of the hypothesis definition, but its meaning is completely different. I advise against using bullet points. If you mention the specific null hypotheses, indicate whether they were rejected. They can probably be omitted.
4.  Eq. (2) should have ":="
5.  L_r has the opposite sign in the equation (2) compared with Figure 3.
6.  Your notation seems to be a bit inconsistent: Your dataset is introduced as bold $\mathbf{D =(X,y)}$ with samples $(\mathbf{X}_i,\mathbf{y}_i)$ and in the following you refer to samples with $(X_i,y_i)$ in the beginning of 3.1 than with $(\mathbf{x},\mathbf{y})$ in eq. (3) and than with $(\mathbf{X},\mathbf{y})$ in eq. (4). I suggest that if you do not refer to the dataset, you just use $(X,y) \in \mathbb{X}\times \mathbb{R}$ for single samples.
7.  Eq. (3) should not have set brackets as the conditions exactly specify some $t \in \mathbb{R}$. Furthermore, both conditions $P((-\infty,t]|x) \geq\tau)$ and $P([t,\infty)|x) \leq1 - \tau\ (\Leftrightarrow P((t,\infty)|x) \leq1 - \tau)$ appear to combine to
$P((-\infty,t]|x) =\tau$ or equivalenty  $P([t,\infty)|x) =1 - \tau$ as $P([t,\infty)|x)+ P((-\infty,t]|x) =1$? Does this make sense and can be interpreted?

---

> ### Author Response · Authors · 2026-02-26
> **Response to Hx61**
>
> We are thankful for Hx61s critical concerns and remarks, and follow up on this reviewers suggested critical changes (CC) with (CC1) a more technical overview of SQRs model implementation in Section 3.3, (CC2) a short discussion on related quantile regression approaches in Section 2.3, focusing on the works included in the evaluation, and (CC3) we implement an additional experiment to assess generalizability to 60th, 70th, 80th quantiles on top of the existing 50th and 90th to better cover the full distribution. These indicate a consistent performance across these levels. We mention(CC4) the runtime limitations in the current implementation in the discussion.
>
> We will adopt the suggested minor changes 1-6 and are thankful for this reviewers’ attention to detail. With respect to minor change 7, the equivalence suggested by this reviewer is correct only for cases with a strictly increasing continuous cumulative distribution function (CDF), but not for cases where the CDF is discontinuous such as those base on a Poisson distribution. The standard definition of the quantile function therefore contains both sides of the quantile and we prefer to keep it in our presentation to align better with the standards accepted in the literature.

---

> > ### Comment · Reviewer_Hx61 · 2026-03-01
> >
> > The authors have addressed the requested changes satisfactorily. Section 3.3 now provides a proper technical description of the SQR implementation, Section 2.3 adds the necessary context on quantile regression, and the additional experiments across τ ∈ {0.6, 0.7, 0.8} and the OOD setting (Table 4) support the generalizability claims. The expanded Appendix E with Figure 6 usefully characterizes when and why SQR underperforms.
> >
> > Two concerns remain before I can recommend acceptance:
> >
> > 1. The general author response incorrectly states that the OOD results appear in "Section 5.3". They are in Section 4 (Table 4). Section 5.3 does not exist. The authors should confirm this is a typo in the response and not indicative of a version mismatch between the submitted paper and the described experiments.
> >
> > 2. The PySR hyperparameters in Appendix D differ substantially between the original and revised submission (niterations: 40 → 900; populations: 15 → 31; parsimony coefficient: 0.0032 → 0.0), yet the results in Table 2 are identical across both versions. These changes are not mentioned anywhere in the author response. The authors should clarify which configuration was actually used to generate the reported results, and confirm that the linked code repository reflects this configuration.

---

> ### Author Response · Authors · 2026-03-01
> **Remaining concerns**
>
> We are thankful of Hx61's assessment and here respond to the novel concerns rightfully raised:
>
> 1. The rebuttal contains a typo in referring to these OOD results. It should have read Table 4 in Section 4 as rightfully pointed out put by this reviewer.
> 2. the new hyperparameter values in Appendix D are correct. The old values produced comparable results (except for runtimes) and were included due to a versioning mistake. We re-ran some single-dataset experiments to validate that the new hyperparameter values are consistent with the results presented earlier and apologize for omitting this change from the author response.

---

### Review · Reviewer_u4MG · 2026-02-11

**Summary Of Contributions:**

# SUMMARY

The authors introduce Symbolic Quantile Regression (SQR), a technique for learning a quantile regression model building on ideas from symbolic regression.  This contribution has two intended benefits: 1) learning models that capture a fuller picture of the conditional output distribution compared to regular regression models, which can be useful in high-stakes applications, and 2) achieving good performance without compromising transparency (as SR learns closed-form expressions by combining basic operations). SQR explicitly optimizes for performance and interpretability. An empirical comparison is carried out on more than 100 datasets, with promising results, and on a aviation case study.

# STRENGTHS/WEAKNESSES

* **Clarity**: good.  The text is well written and easy to follow.
  - Minor CON: One complaint I have is that there's no guarantee that a purely statistical learning approach can uncover the "phenomena underlying the data" - that's more likely a causal problem, requiring counterfactual data or further assumptions for proper identification. In this sense, I find the "discovery use case" not entirely well justified.
  - Minor CON: Steinward and Christmann should be referenced as early as possible in Section 3, for fairness. Otherwise it's unclear where the equations (2-4) come from.
  - Nitpick: I would appreciate if the authors could move the abbreviations in Eq 7 and 8 into the text.
  - Nitpick: I don't think $y$ should be bold in the equations - it is not a vector, right?
  - Nitpick: One citation did not compile properly (end of Section 4).
  - Nitpick: page 9: "expressions. we" - spurious full stop.
* **Significance**: SQR has convincing applications in a number of areas, as documented in the introduction.
* **Novelty**: SQR combines existing ideas and tools: the pinball loss, PySR. As such, novelty is not exceptional.
* **Quality**:
  - PRO: The evaluation considers 122 datasets (real-world and synthetic).
  - PRO: The choice of metrics is good, as is the choice of running 5-fold cross validation with statistical tests.
  - PRO: The choice of competitors is also reasonable.
  - PRO: Hyperparameters for the competitors are chosen with Optuna, which is good. More generally, hyperparameter choices are provided in the appendix.
  - PRO: The aggregated results look promising.
  - PRO: The authors are upfront about the runtime of their approach, which is substantially worse than those of the competitors.
  - Major CON: [R1] I don't quite understand how to interpret the fit reported in Section 5.2: I would replace the pinball loss (currently sitting at 1746.29) with its normalized variant. If it is already normalized, the authors should say so (or simply call it Quantile Loss). The empirical coverage looks good (close to 50% for the 0.5 model and to 90% for the 0.9 model), but overall I don't quite understand whether the model being discussed "truly" fits the data. I am worried the model might be over-regularized to facilitate interpretation at the expense of data fit. Please clarify.
  - Medium CON: [R2] The aggregated results in Table 2 are not discussed in much detail, and the detailed results in Appendix E come with no discussion at all. Are there any cases in which SQR underperforms and, if so, why? More generally, I would appreciate if the authors could grouping datasets into clusters for the purpose of understanding when and why SQR works as well as it does (and when it doesn't).
  - SUGGESTION/QUESTION: would it make sense to summarize the results in Appendix E using a couple of histograms or scatterplots? (e.g., coordinates could indicate Quantile Loss vs Parsimony with color used for method). These could provide an at-a-glance idea of how the various competitors perform.
  - QUESTION: Will the code be released?

**Audience:**

Yes

**Audience Explanation:**

Both symbolic regression and quantile regression fall within the interests of TMLR's readers.

**Broader Impact Concerns:**

Nothing of note. I expect SQR to raise essentially the same concerns as any other regression technique.

**Claims And Evidence:**

Yes

**Claims Explanation:**

The main claims are as follows:
- SQR can predict conditional quantiles. I agree this is what it is designed to do. The results are also encouraging.
- SQR outperforms transparent models - yes, see Table 2 and Appendix E.
- SQR performs comparably to the black-box baseline - yes, same.
- SQR does not compromise transparency - yes, same, as measured by "parsimony".
In general, the experimental setup and the algorithmic design well reflects the message.

**Requested Changes:**

See R1 and R2 above.

---

> ### Author Response · Authors · 2026-02-26
> **Response to u4MG**
>
> We appreciate u4MG’s review and critical concerns.
>
> Regarding the first Minor CON on the __understanding use-case__, we agree that fully automatically uncovering phenomena is not a problem to which SQR is suited when used in isolation. In the introduction we therefore use the phrasing “Discovering an expression that is both predictive and interpretable not only enhances understanding of the phenomena underlying the data [...]” in the original draft. We have added a sentence echoing this stance in the discussion “While fully uncovering the phenomena is a causal problem which may require counterfactual data and/or further assumptions to address fully automatically, our SQR approach can enhance understanding of the phenomena at hand based on a given set of data.”
>
> Regarding the second Minor CON, we have included references to seminal works on QR by Koenker & Basset Jr, and Steinwart & Christmann to the first mention of QR in the introduction. We have addressed all nitpicks as well and are thankful for u4MGs attention to detail.
>
> Regarding Major CON [R1] we originally incorporated the unnormalized loss to allow domain experts to assess how useful the approach is in determining the fuel amount to load in practice. However, after this reviewers’ concern, we fully realize and agree that the unnormalized loss is not interesting or interpretable for the general audience. Ideally, we would provide both the original score and a score normalized on the actual data. However, providing both original and normalized scores would require sharing values of data we are not permitted to disclose by our industrial partner. We therefore suggest to use approximate minimum and maximum values of 12.000 and 80.000kg of fuel based on [0] which leads to an approximate normalized quantile loss of 0.03, which is in line with the main results in Table 2 and indicates a good fit when we incorporate the empirical coverage of 0.91 (e.g. an absolute coverage error of 0.01).
>
> We thank the reviewer for their concern Medium CON[R2] and the suggestion to better present the additional results in Appendix E. Following this suggestion, we have added boxplots to Appendix E that visualize the results across dataset characteristics. These characteristics include dataset size $n$, dimensionality $d$, and the number of categorical features. These plots illustrate that SQR tends to maintain a consistently low complexity score, largely independent of dataset size $n$ or dimensionality $d$. However, predictive performance for SQR degrades for large datasets of low dimensionality, in which case SQRs preference for interpretable solutions interferes with obtaining high predictive performance.
>
> On datasets with a large number of observations (e.g., ID 344 with $n=40,768$), models like QDT and LGBM tend to increase their complexity to capture detailed, non-linear patterns (e.g., QDT parsimony > 12,000). SQR's parsimony remains comparatively low (e.g., 11.60), which can act as a capacity bottleneck. As a result, SQR may underfit the available data, leading to a higher Quantile Loss than the tree-based models.
>
> Linear Quantile Regression (LQR) scales its complexity directly with the number of features (e.g., ID 505 with $d=124$ results in an LQR parsimony of 124). In these settings, LQR is more prone to overfitting, which can lead to higher Absolute Coverage Errors, particularly at extreme quantiles ($\tau=.9$). SQR's tendency to select simpler expressions acts as a form of regularization, which appears to help mitigate overfitting in the high-dimensional regime and at the distribution tails at limited cost in terms of predictive accuracy.
> We have added a new subsection to our results to discuss how this parsimony behavior influences SQR's relative performance.
>
> In our discussion of the analysis, we maintain caution regarding the ability to predict performance of any predictive modeling approach for a particular dataset and therefore opt for a conservative interpretation of these results. We believe these additions provide a more balanced view of the results in Tables 2 and 13, offering some guidance on the applicability of SQR.
>
> All code for the main results will be released with a permissive open access license.
>
> [0] Kühn, M., & Scholz, D. (2023, September 19–21). Fuel consumption of the 50 most used passenger aircraft [Poster presentation]. Deutscher Luft- und Raumfahrtkongress 2023, Stuttgart, Germany. https://doi.org/10.48441/4427.1045

---

### Author Response · Authors · 2026-02-26
**General response to first round of reviews**

We would like to thank the Action Editor and the reviewers for their time, thoughtful feedback, and constructive criticism of our manuscript. We are highly encouraged by the reviewers' appreciation of the relevance of this work to the TMLR audience, the thoroughness of our empirical evaluation across 122 datasets, and the novelty of the aviation case study.

We think that the detailed comments have been instrumental in improving the quality, clarity, and rigor of our paper. In response to the reviews, we have made several substantial additions and revisions to the manuscript:
* _Extrapolation and Generalization (Reviewer 48Jj)_: To address concerns regarding how well SQR extrapolates beyond the support of the training data, we designed and executed a new large-scale out-of-distribution experiment. We split the data at the 90th quantile of the most informative feature. The results (now in Section ~~5.3~~ 3 and Table 4) demonstrate that SQR maintains its position as the best-performing transparent model with only a moderate predictive performance penalty, showcasing its robustness.
* _Broader Quantile Evaluation (Reviewer Hx61)_: We expanded our experimental scope beyond the 50th and 90th quantiles to include the 60th, 70th, and 80th quantiles. This confirms SQR's consistent performance across the full distribution and strengthens our core claims.
* _Contextualizing Data Fit and the Aviation Use Case (Reviewers u4MG & 48Jj)_: We clarified the interpretation of the pinball loss in our real-world use case by providing an approximate normalized quantile loss, restricted only by the data-sharing limitations of our industrial partner. Furthermore, we have shifted our presentation of results to be suitable for a general audience, along with some domain-specific details to explain the additive relationship of the discovered quadratic ASF term.
* _Implementation and Related Work (Reviewer Hx61)_: We enhanced the manuscript's completeness by adding a technical overview of SQR's model implementation in Section 3 and incorporating a broader discussion of related interpretable quantile regression approaches in Section 2.
* _Performance Analysis and Visualizations (Reviewer u4MG)_:To address the request for deeper insights into the aggregated results (Table 2 and Appendix E) and to clarify the conditions under which SQR might underperform, we have expanded our analysis with a visualization. SQR generally maintains a low parsimony score regardless of dataset size or dimensionality. Its strong bias toward low-complexity models can be beneficial in high-dimensional or low-sample settings, as well as for extreme quantiles ($\tau=.9$), where less constrained models like LQR may overfit. However, this same characteristic can limit SQR's capacity on datasets with large sample sizes. In such cases (large $n$, low $d$), SQR may underfit compared to models like QDT or LGBM, which can scale their predictive performance better with dataset size albeit at the cost of high complexity for QDT. To visualize these trade-offs, we have added new boxplots to Appendix E, and added paragraphs to the results and discussion section describing these insights.

Finally, we have addressed all minor concerns, mathematical nitpicks, and typographical errors pointed out by the reviewers. We will also be releasing all code with a permissive open-access license to facilitate reproducibility and future research.

We believe these revisions comprehensively address the concerns raised and significantly elevate the manuscript and want to thank all reviewers again for their excellent suggestions.

Sincerely, The Authors

---

### Decision · Action_Editor_WHyc · 2026-04-08

**Recommendation:** Accept as is

**Audience:**

Yes

**Audience Explanation:**

Both symbolic regression and quantile regression are well aligned with the interests of the TMLR readership.

**Claims And Evidence:**

Yes

**Claims Explanation:**

The paper introduces Symbolic Quantile Regression (SQR) with four main claims: SQR successfully predicts conditional quantiles, SQR outperforms other transparent models, SQR performs comparably to black-box models, and SQR maintains transparency as measured by parsimony following established conventions (Petersen et al., 2021; 2020). Claims are empirically validated across a large benchmark of 122 regression datasets and in a case study from the commercial aviation domain.

Symbolic Quantile Regression applies existing tools, PySR (Cranmer, 2023) and the pinball loss (Koenker & Bassett Jr, 1978). While the contribution is primarily empirical rather than theoretical, the strength and breadth of the empirical validation make the claims convincing.